# SemiHVision: Enhancing Medical Multimodal Models with a Semi-Human Annotated Dataset and Fine-Tuned Instruction Generation

## Abstract

Many medical MLLMs post strong scores on curated VQA-style benchmarks yet still struggle on real clinical questions because their training/supervision expose them to too little clinically grounded knowledge and prevailing benchmarks contain too few diagnostic-reasoning Q&A items. We introduce **SemiHVision**, a semi-human–validated multimodal instruction dataset built with a multimodal retriever; to our knowledge, this is the first dataset to leverage a unified image–text retriever to integrate real-world clinical information into data construction, thereby strengthening models' clinical diagnostic reasoning. Our pipeline retrieves image- and context-relevant evidence and performs retrieval-augmented synthesis to produce clinically grounded instruction Q&A and captions across major modalities (X-ray, CT, MRI, ultrasound, histopathology), while standardizing heterogeneous annotations into a training-ready schema. For model fine-tuning, we train **SemiHVision-8B-AN**, surpassing public medical models like HuatuoGPT-Vision-34B (79.0% vs. 66.7%) and private general models like Claude3-Opus (55.7%) on standard benchmarks (SLAKE, VQA-RAD). On the JAMA Clinical Challenge—a benchmark that directly probes diagnostic reasoning aligned with clinical practice—we evaluate SemiHVision-AN and it achieves a GPT-4 rubric score of 1.29, exceeding HuatuoGPT-Vision-34B (1.13) and Claude3-Opus (1.17), indicating the effectiveness of SemihVision datasets [1].

## 1 Introduction

Large Multimodal Models (LMMs) show strong potential for general medical AI (Yan et al., 2023; Liu et al., 2024b; Jin et al., 2024; Li et al., 2024; Chen et al., 2024b), and recent works adapt general models to medicine by fine-tuning on PMC image–text corpora (e.g., LLaVA-Med, HuatuoGPT-Vision, MedTrinity) (Li et al., 2024; Chen et al., 2024b; Xie et al., 2024). Despite this promise, datasets remain the bottleneck: current collections diverge from real diagnostic workflows and multi-view evidence, preventing medical MLLMs from acquiring robust, clinically grounded reasoning.

Prevailing medical-MLLM datasets drive the core limitations: **(L1) lack of end-to-end, real diagnostic workflows**—PubMed/PMC image–caption pairs or synthetic captions dominate, and datasets rarely couple complete imaging studies with patient history and expert-authored reports (Four Stages: evidence → findings → discussion → diagnosis); datasets seldom link region-level evidence across views/series or longitudinally, so supervision does not cover the full diagnostic workflow (see Fig. 4) (Bustos et al., 2020; Irvin et al., 2019; Johnson et al., 2019; Ikezogwo et al., 2024). **(L2) lack of multi-view evidence**—datasets typically compress a clinical case into a single global caption of an isolated image—often model-generated—while omitting non-imaging clinical evidence (e.g., laboratory tests, vital signs, medications, prior history) and provide no structured linkage between imaging findings and these signals; thus, supervision skews toward caption-/QA-style associations rather than integrated diagnostic synthesis grounded jointly in imaging and clinical data (Xie et al., 2024; Pal et al., 2023; Dorfner et al., 2024). Consequently, such datasets yield models that struggle

[1]We will share our code and datasets: https://anonymous.4open.science/r/PMC-Cambrian-CBC2/README.md

---

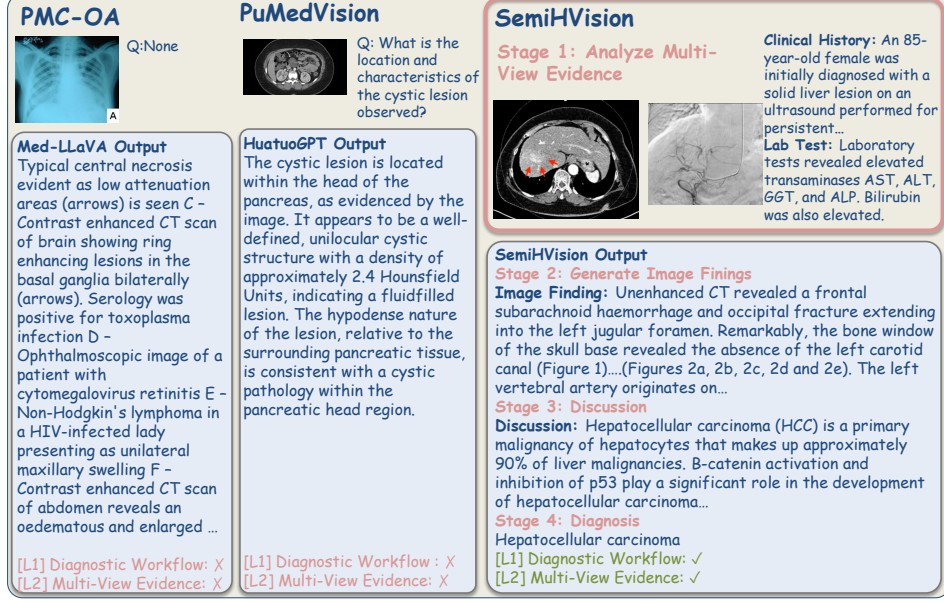

Figure 1: **Left (PMC–OA).** Items are isolated images with generic, list–style captions and no task prompt. There is no diagnostic workflow (evidence → findings → differential → impression/plan) and no multi-view linkage across images from the same case; models learn caption recall rather than case reasoning. **Middle (PubMedVision)**. Adds single–image Q&A, but supervision still centers on one view at a time with no study structure, ROI provenance, or cross-view/multi-modal alignment; the diagnostic pipeline remains unsupervised. **Right (SemiHVision).** Each case is organized by **staged diagnostic workflow**: Stage 1 analyzes **multi-view/multi-modal evidence** from the same study; Stage 2 produces structured image findings with ROI grounding; Stage 3 writes the discussion (linking evidence to differentials); Stage 4 states the diagnosis/next step. Clinical history and labs are integrated and claims are attributed to specific views/ROIs, enabling end-to-end, evidence-based supervision.

with real-world clinical diagnosis, and the deficiency remains largely undetected—highlighting the urgent need for data and benchmark improvement (Liu et al., 2021; He et al., 2020; Lau et al., 2018).

To address these limitations, we curate SemiHVision—a semi-human–validated instruction corpus that integrates real, de-identified clinical datasets rather than synthetic caption pairs. Concretely, we link complete imaging studies to authentic clinical context (expert reports, laboratory tests, vitals, medications, and relevant history), harmonize modalities and series structure (X-ray, CT, MRI, ultrasound, histopathology), and normalize report fields into process-centric supervision. The resulting instruction Q&A and evidence-linked captions are produced with targeted automatic augmentation and expert verification, yielding an evidence-grounded, training-ready schema centered on diagnostic reasoning; when helpful, a multimodality lightweight retriever adds guideline/textbook references. The pipeline is shown in the Figure 2. To valid our pipeline could work, we train LLM on our datasets. Then we validate our model on widely used benchmarks such as SLAKE and VQA-RAD, where it achieves state-of-the-art performance, outperforming both public medical MLLMs (e.g., HuatuoGPT-Vision-34B) and strong general models. To address the lack of evaluation targeting fine-grained diagnostic reasoning limitation—we further introduce the JAMA Clinical Challenge, a benchmark curated from real clinical case vignettes designed to test problem framing, differential diagnosis, evidence-based reasoning, and final judgment. We complement these evaluations with rubric-based metrics (e.g., GPT-4 grading and human evaluation) to enable fair and nuanced comparisons between medical and general MLLMs.

## 2 RELATED WORK

### 2.1 EXISTING MULTIMODAL MEDICAL DATASETS

Medical multimodal datasets have evolved from report–paired corpora such as MIMIC-CXR-JPG (Johnson et al., 2019) to large image–caption collections like PMC-OA (Lin et al., 2023),

case/VQA resources (PMC-CaseReport, PMC-VQA) (Wu et al., 2023; Zhang et al., 2023), instruction–style sets from PubMed imagery (LLaVA-Med VQA, PubMedVision) (Li et al., 2024; Chen et al., 2024b), structured annotation efforts (RadGenome-Chest CT) (Zhang et al., 2024), and multi-granular pipelines (MedTrinity) (Xie et al., 2024). Despite progress, prior corpora commonly lack of end-to-end, real diagnostic workflows. Most corpora reduce supervision to single captions or short Q&A and omit the stepwise pipeline (history → findings → differential → impression/plan) and provenance, so models practice recall rather than executing the clinical workflow; and lack of multi-view evidence—images from the same case are treated independently, with little linkage across views/series/modalities (e.g., AP/LAT, CT slices, MRI sequences), preventing synthesis of corroborating/contradictory evidence. In contrast, SemiHVision is an expert-in-the-loop, multimodal-retriever–grounded corpus that restores L1 by supervising staged reports and decision targets and addresses L2 by aligning views/slices/modalities within each case and context-linked attributions, yielding instruction signals centered on evidence-based diagnostic reasoning across major modalities.

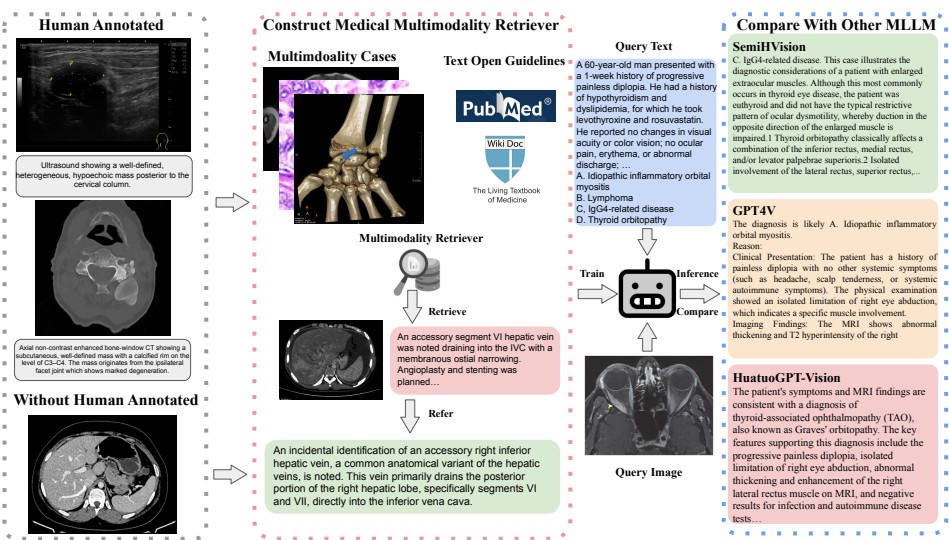

Figure 2: **SemiHVision curation pipeline.** We target two data gaps—(i) lack of end-to-end, clinical diagnosis supervision and (ii) neglect of non-imaging clinical evidence. Human-annotated branch: clinicians annotate image findings and link them to clinical history and labs/vitals/medications, aligning report structure (findings → differentials → impression/plan). GPT-4o then reformats these into instruction-style Q&A and evidence-linked captions without altering the expert content. Unannotated branch: for studies without labels, a lightweight retriever surfaces openguidelines snippets (Chen et al., 2023) and similar cases; GPT-4o drafts instructions/captions conditioned on this context, followed by expert screening and edits. Both branches are standardized into a training-ready schema that preserves (a) study-level clinical context, (b) region-level imaging evidence, and (c) stepwise diagnostic reasoning targets, yielding supervision aligned with real clinical diagnosis rather than caption-style recall.

## 2.2 Medical Multimodal Model

In recent years, several efforts have fine-tuned general-purpose multimodal models on medical data, yielding promising results. For example, Med-Flamingo (Moor et al., 2023) adapted OpenFlamingo-9B (Chen et al., 2024a) with a small-scale medical corpus, and Med-PaLM (Tu et al., 2024) extended PaLM-E (Driess et al., 2023) using one million medical image–text pairs. Similarly, models like LLaVA-Med, Med-Gemini (Saab et al., 2024), and HuatuoGPT Vision have utilized instruction tuning over curated PubMed-derived datasets for medical QA. However, these efforts primarily rely on image–caption pairs or short-form Q&A extracted from biomedical literature (e.g., PubMed), and therefore lack exposure to the complex, multi-step reasoning required in real-world clinical scenarios. In contrast, we train SemiHVision-8B-AN model on our SemiHVision corpus, which is explicitly grounded in clinical guidelines and case-based supervision. As a result, our model demon-

strates stronger capabilities in end-to-end clinical reasoning, including patient history interpretation, differential diagnosis, evidence justification, and final decision-making.

## 3 SEMIHVISION

SemiHVision explicitly tackles (L1) the lack of end-to-end, real diagnostic workflows and (L2) the lack of multi-view/multi-modal evidence by designing multi-stage instruction fine-tuning data directly from clinical cases. Each case is organized along the diagnosis workflow and yields supervision for detection/localization → evidence attribution → diagnosis/next step, while evidence from the same case is linked across views and modalities with ROI grounding and clinical context. This case-centric, study-level construction both supervises the full diagnostic workflow (resolving L1) and aligns cross-view/cross-modal signals needed for accurate synthesis (resolving L2).

### 3.1 DATA COLLECTION

**Data Source and Image Selection Strategy** To endow the model with the missing **L2** capability, we curate complete, multi-view/multi-modal datasets, preserve series/view structure, and deliberately balance coverage across CT/X-ray/MRI/US/histopathology so the model can learn cross-view correspondences and case-level synthesis. For pretraining, we filter PubMed-derived items (25M → 14M after removing corrupted/short texts) and purge non-medical PMC items using a lightweight classifier, then *rebalance toward underrepresented, clinically critical modalities* (MRI, X-ray) and retain series/view structure (AP/LAT, CT slices, MRI sequences). For 3D studies, we use provided slice IDs and evenly sample additional slices, capping each study at $\leq 20$ 2D slices to preserve *intra-study continuity* without overwhelming redundancy. This case/study-centric sampling preserves cross-view correspondences and ROI continuity needed to learn multi-view fusion (details in Appendix A.8, Table 8).

**Human Annotated Workflow** To supervise the end-to-end diagnostic workflow (**L1**), we prioritize expert–labeled sources (e.g., Eurorad, Radiopaedia) that mirror real workflows (history → findings → differential → impression/plan). Because raw annotations vary in length and style, we standardize them into consistent workflow fields and align sentences to views/ROIs. For lengthy reports (e.g., Eurorad), we decompose into (i) per-image findings, (ii) study-level synthesis, and (iii) discussion/decision text, regenerating only format (not content) to ensure consistency while preserving expert intent. The result is study-level supervision that simultaneously (i) teaches stepwise reasoning and decisions (addresses L1) and (ii) ties claims to specific views/slices and clinical context (supports L2). Further details appear in Appendix A.9.

### 3.2 DATA CONSTRUCTION PIPELINE

Guided by the above limitations, we construct a clinically grounded multimodal corpus from two sources—human-annotated clinical cases and unannotated medical images—so that training targets preserve study structure and fuse imaging with non-imaging clinical evidence.

**Stage I: Clinical curation and indexing.** We curate *complete imaging studies* with accompanying clinical context from Eurorad and Radiopaedia (reports, case descriptions, and region-of-interest (ROI) hints when available), and build a guideline repository from OpenGuidelines (Chen et al., 2023). Each study retains its series/view organization (e.g., multi-view radiographs, CT/MRI slices) and is linked, when available, to patient-level variables (history, labs, vitals, medications). We normalize report sections into a consistent template (*findings*, *differentials*, *impression/plan*), harmonize modality/view tags, and de-duplicate near-identical cases. A lightweight image+text retriever (UniIR with fusion scoring) indexes both the guideline corpus and the image–report collection, enabling retrieval of authoritative guidance and closely related cases for studies lacking annotations. This preserves end-to-end diagnostic context rather than reducing supervision to single-image captions.

**Stage II: Multi-View Evidence Linked.** For *human-annotated* cases, we restructure existing reports into a process-centric template (evidence → Image Findings → Diagnosis) and align sentence spans to ROIs; clinical signals (history, labs, vitals, medications) are propagated and explicitly referenced where they inform the differential. For unannotated images, the retriever surfaces a small

set of supportive items (typically $k=4$; at least one guideline plus similar cases), which serve as context for GPT-4o to draft evidence-linked captions and provisional reports conditioned on study structure and ROI cues. Drafts undergo expert screening/edits to remove unsupported statements, enforce explicit ties between textual claims and image evidence (with ROI references), standardize terminology (e.g., laterality, anatomy), and ensure that non-imaging clinical signals are correctly integrated. This converts both branches into supervision that teaches how localized visual evidence and clinical context jointly shape the differential and impression.

**Stage III: Diagnosis Workflow Construction.** From the curated reports and captions, we programmatically form instruction–response pairs that supervise the diagnostic workflow end-to-end: (i) Detection/Localization—identify and localize salient findings with explicit slice/view/ROI references; (ii) Evidence Attribution—explain differentials by citing the supporting image regions and pertinent clinical signals; (iii) Diagnosis & Next Step—state a working diagnosis and propose an appropriate next diagnostic action. To reduce shortcut learning, we introduce normal and negative constructions with clear purpose and provenance: normal controls (studies without acute findings, requiring a justified "no acute finding" answer), absent-lesion distractors (plausible pathologies drawn from guidelines/similar cases but not present in the current study, used to test evidence checking), and near-miss distractors (findings that occur in anatomically adjacent regions or on alternate views/slices to test precise localization). For each negative, responses must state why the distractor is not supported (e.g., incorrect region, contradictory clinical labs), which trains the model to verify rather than recall. We balance positive/negative instances per study (e.g., 1:1–1:2 depending on modality) and label each item with machine-readable fields (study id, modality, series/view, ROI refs, clinical signals, findings, differentials, diagnosis, next step, evidence citations) to produce a training-ready schema consistent across views/series and modalities. The overall process is illustrated in Figure 2. Finally, we conducted a human evaluation of data quality. Finally, we conducted a human evaluation of data quality. Three physicians (each with 10+ years of clinical experience) independently reviewed 100 randomly sampled cases, checking whether the synthesized constructions matched the original cases; 95% were fully consistent, with an inter-rater agreement score of 0.90.

## 3.3 DATA FEATURE ANALYSIS

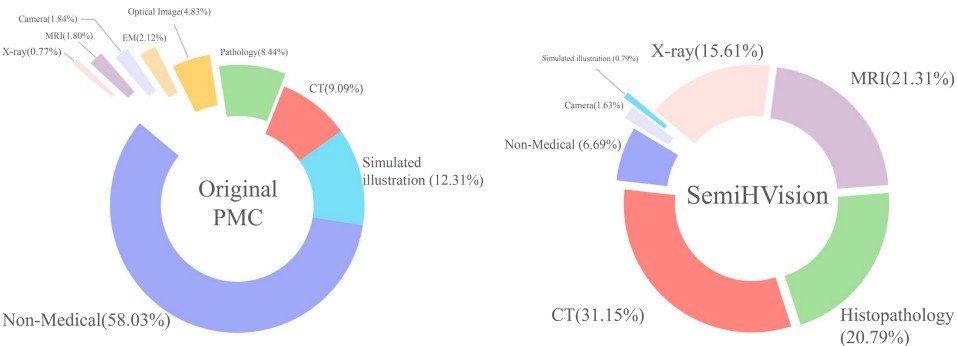

Figure 3: A comparative distribution of image modalities between the original PMC dataset and the SemiHVision dataset. The original PMC dataset contains a significant portion of non-medical content (58.03%), with a relatively lower representation of key medical imaging modalities like MRI (1.80%) and X-ray (0.77%). In contrast, the SemiHVision dataset demonstrates a more balanced distribution, with a substantial increase in clinically relevant modalities such as CT (31.15%), MRI (21.31%), and X-ray (15.61%), while minimizing the presence of non-medical images (6.69%).

Unlike traditional methods for generating instruction datasets, we collected a broader range of human-annotated data across multiple modalities. We conducted a distribution analysis on randomly sampled 200k entries from both the original PMC and SemiHVision datasets. Expert annotators classified the images into categories such as X-ray, DSA, CT, MR, PET/SPECT, Ultrasound, Histopathology, and others. Additionally, we employed GPT-4o for image classification, and to ensure accuracy, a random sample of 100 images was reviewed by human experts, yielding a classification accuracy of 73%. We focused on analyzing higher-frequency modalities, as depicted in Figure 3. The analysis

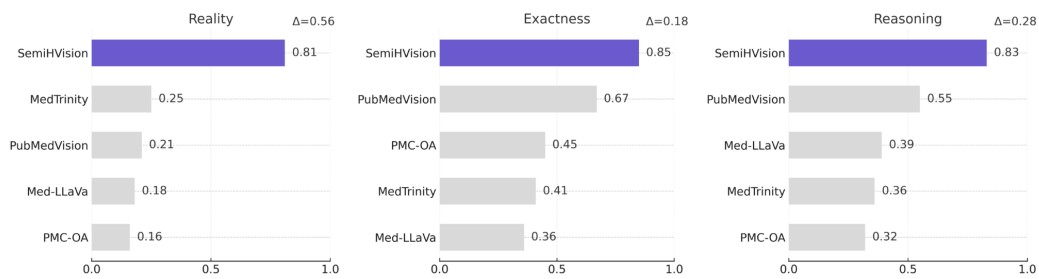

Figure 4: Compared with legacy medical QA datasets, which mainly provide image captions or caption-derived Q&A without full case context (e.g., PMC-OA and MedTrinity list images with descriptive text but no question; PubMedVision offers PubMed-style Q&A), SemiHVision pairs images with clinical history, imaging findings, differential cues, and diagnosis prompts that mirror real workflows. We quantify *clinical-scene fidelity* using a five-level rubric (1–5; poor→excellent). First, automatic screening: Qwen2.5-VL scores up to 1M samples per dataset by assigning log-probabilities to labels 1–5 and converting them into expected rubric scores; we rank items and keep the top 1k per dataset. Second, three physicians independently rate the retained datasets along *Reality*, *Exactness*, and *Reasoning*; scores are mapped to five grades and then normalized to $[0, 1]$, after which we report per-dataset means. The figure shows three panels (one per axis) as $\Delta$–lollipop plots: each marker's horizontal position encodes the normalized mean; $\Delta$ annotates the gap between the best and second-best. Result: current medical–QA datasets remain far from real clinical scenarios on all three axes, while SemiHVision achieves the highest clinical-scene fidelity.

revealed that non-medical images constitute a significant portion of the original PMC dataset, with simulated illustrations like statistical charts being the second largest category. In contrast, clinically critical modalities like CT, MRI, and X-ray were significantly underrepresented, highlighting the scarcity of these essential medical images in the PMC dataset. Despite prior filtering efforts, the low representation of modalities like MRI and X-ray means the final dataset still lacks sufficient numbers of these images. For the SemiHVision dataset, we performed a similar sampling and distribution analysis. Unlike the PMC dataset, not all entries were classified using GPT-4o, as some, such as those from Quilt-1M, were already pre-labeled. The resulting distribution demonstrates that SemiHVision contains a more balanced representation of clinically relevant modalities. Notably, modalities underrepresented in the PMC dataset, such as MRI and X-ray, have a much higher proportion in SemiHVision, ensuring more comprehensive coverage of medical knowledge essential for model training and expanding the scope of medical expertise.

## 4 EXPERIMENT SETTINGS

We adopt a two-step protocol. **Step 1 (L1)**: train SemiHVision-8B with workflow-supervised SFT to teach the full clinical pipeline (history → findings → differential → impression/plan). We then evaluate both traditional medical VQA (SLAKE, VQA-RAD, PathVQA, PMC-VQA) and JAMA to verify that learning the workflow yields broad gains across standard tasks and improves diagnosis structure on case problems. **Step 2 (L2)**: after annealing on study-level, multi-view/multi-modal cases to teach cross-view synthesis. We re-evaluate, with emphasis on JAMA's rubric (Key Points/Inference/Evidence) and cross-view consistency checks, to test whether the model now uses multi-view evidence to handle real-world clinical diagnosis.

### 4.1 TRAINING EXPERIMENT SETUP

During the training of SemiHVision-8B, we employed a two-stage process. First, we filtered the original PMC dataset by removing captions with fewer than 20 words, yielding a final dataset of 14 million samples. We then pre-trained the model on this refined dataset using a learning rate of 1e-4 and an image token length of 512. DeepSpeed Stage 2 was utilized, with a batch size of 8 and a gradient accumulation step of 6. During this stage, we focused solely on training the adapter while

| Model | VQA-RAD | SLAKE | PathVQA | PMC-VQA | Avg. |
|---|---|---|---|---|---|
| GPT-4o-mini | 45.9 | 59.0 | 37.9 | 33.3 | 44.0 |
| Claude3-Opus | 52.5 | 55.2 | 54.3 | 60.7 | 55.7 |
| Med-Flamingo | 45.4 | 43.5 | 54.7 | 23.3 | 41.7 |
| RadFM | 50.6 | 34.6 | 38.7 | 25.9 | 37.5 |
| LLaVA-Med-7B | 51.4 | 48.6 | 56.8 | 24.7 | 45.4 |
| Qwen-VL-Chat | 47.0 | 56.0 | 55.1 | 36.6 | 48.9 |
| Yi-VL-34B | 53.0 | 58.9 | 47.3 | 39.5 | 49.7 |
| LLaVA-7B | 52.6 | 57.9 | 47.9 | 35.5 | 48.5 |
| LLaVA-13B | 55.8 | 58.9 | 51.9 | 36.6 | 50.8 |
| LLaVA-34B | 58.6 | 67.3 | 59.1 | 44.4 | 57.4 |
| LLaVA-8B | 54.2 | 59.4 | 54.1 | 36.4 | 51.0 |
| + LLaVA_Med | 60.2 | 61.2 | 54.5 | 46.6 | 55.6 |
| + PubMedVision | 63.8 | 74.5 | 59.9 | 52.7 | 62.7 |
| HuatuoGPT-Vision-34B | 68.1 | 76.9 | 63.5 | 58.2 | 66.7 |
| **Our Model** | | | | | |
| SemiHVision-8B-20M | 67.8 | 76.1 | 57.8 | 53.6 | 63.8 |
| SemiHVision-8B | 69.2 | 77.2 | 63.6 | 58.4 | 67.1 |
| SemiHVision-8B-Mix | **74.2** | **81.3** | **76.3** | **59.1** | **72.2** |
| SemiHVision-8B-AN | **86.1** | **87.7** | **80.4** | **61.9** | **79.0** |

Table 1: Performance comparison of various models on medical VQA benchmarks (VQA-RAD, SLAKE, PathVQA, PMC-VQA) with average scores is presented. SemiHVision-8B-20M refers to the model trained using all slices from the 3D dataset. SemiHVision-8B prioritizes human-annotated slices and selectively sampled portions for training, using GPT-4o-generated synthetic data. SemiHVision-8B-Mix is trained by combining both the human-annotated datasets and the GPT-4o-generated synthetic datasets. SemiHVision-8B-AN is the result after annealing on human-annotated datasets based on SemiHVision-8B.

freezing the other model components. The pre-training phase ran on four H100 GPUs for 420 hours. This stage provides generic vision–language grounding prior to task-specific supervision.

In the fine-tuning phase, we used the SemiHVision dataset with a learning rate of 2e-5, while keeping the DeepSpeed Stage 2 configuration, with a batch size of 6 and a gradient accumulation step of 6. Unlike the pre-training phase, the full model parameters were trained. This fine-tuning process was conducted on 8 H100 GPUs for 90 hours. For instruction tuning, we divided the process into two phases: standard instruction tuning and the Annealing phase which is the same as Llama3 (Dubey et al., 2024). The learning rate in Annealing phase is 1e-5. During the instruction tuning phase, we used non-human-annotated data, primarily GPT-4o-generated synthetic data. In the Annealing phase, we focused on human-annotated data, where GPT-4o applied further augmentation to enhance the dataset (The details are shown in Appendix A.3).

## 4.2 AUTOMATIC EVALUATION PIPELINE

We evaluate on both traditional medical VQA benchmarks and a case-based JAMA benchmark (Appendix A.4). Because surface-similarity metrics (e.g., F1/ROUGE) are ill-suited to clinical reasoning, our pipeline uses two stricter measures: *UMLS-F1* (concept overlap via SciSpacy/UMLS; Appendix A.7) and a blinded *GPT-4o rubric score*. The rubric assesses fine-grained diagnostic ability along three doctor-designed axes—**Key Points** (coverage of critical clinical elements in the reference), **Inference** (correctness and completeness of the stepwise diagnostic path), and **Evidence** (whether claims are grounded in specific findings, imaging views/ROIs, or clinical signals). To reduce stylistic bias, model outputs are style-normalized before judging, and the judge sees only extracted gold summaries of *Key Points/Inference/Evidence*, not the full reference text. Concretely, on Medical VQA (SLAKE, VQA-RAD, PathVQA, PMC-VQA) we report **accuracy** to test whether learning the end-to-end workflow (**L1**) yields broad gains on standard tasks; on JAMA we report (i) close-ended **accuracy** where applicable, (ii) **UMLS-F1**, and (iii) the blinded rubric score, which explicitly stresses multi-view evidence use and attribution, thereby probing **L2**.

# 5 RESULTS

## 5.1 ADDRESSING L1 RAISES PERFORMANCE ON TRADITIONAL BENCHMARK

**Results on Traditional Benchmark.** Table 1 shows that SemiHVision models fine-tuned on GPT-4o synthetic data significantly outperform both general-purpose and medical-specific models on standard medical VQA benchmarks: SemiHVision-8B reaches an average **67.1%**, surpassing the much larger HuatuoGPT-Vision-34B (**66.7%**) and exceeding a similar-sized LLaVA-8B trained on PubMedVision by **+4.4%**. Further, when we *anneal* by continuing training on human-annotated diagnostic data, SemiHVision-8B-AN achieves an outstanding **79.0%**, outperforming SemiHVision-8B-Mix (**72.2%**) and beating HuatuoGPT-Vision-34B by **18.4%**. It also exceeds private models Claude3-Opus (**55.7%**) and GPT-4o-mini (**44.0%**). These results indicate that—notwithstanding parameter count—our study-aware, clinically grounded supervision delivers larger gains on recall-heavy VQA metrics than caption-centric pretraining alone, and annealing on human-annotated cases further amplifies these gains.

**Why fixing L1 helps on traditional VQA.** L1 targets the lack of end-to-end, clinically workflow supervision. By preserving study structure (views/series), tying text to report fields (findings, differentials, impression), and injecting clinically grounded targets via annealing, supervision shifts from single-image captions to case-level, evidence-aware signals. Although traditional VQA benchmarks primarily reward knowledge recall rather than full diagnostic workflow, this end-to-end, study-aware supervision increases structured fact density and reduces spurious shortcuts, which directly maps to higher answer accuracy on recall-style questions. Hence, addressing L1—first through curated, report-anchored pretraining (yielding **67.1%**; **+4.4%** over PubMedVision and even above the **66.7%** larger model) and then through annealing with human-annotated diagnostic data (up to **79.0%**, **+18.4%** over Huatuo-34B)—systematically raises traditional benchmark performance without relying on parameter scale.

**Annealing and overall lift.** To demonstrate the importance of annealing, we trained two models: SemiHVision-8B-Mix, which mixes GPT-4o synthetic data and human-annotated data, and SemiHVision-8B-AN, which is first trained on GPT-4o synthetic data and then annealed on human-annotated data. SemiHVision-8B-AN achieves an outstanding **79.0%** average accuracy, surpassing SemiHVision-8B-Mix (**72.2%**) and outperforming HuatuoGPT-Vision-34B by **18.4%**. Compared to private models like Claude3-Opus (**55.7%**) and GPT-4o-mini (**44.0%**), SemiHVision-8B-AN consistently excels across benchmarks, underscoring that addressing L1 (end-to-end, clinically grounded supervision of studies rather than single captions) systematically raises traditional recall-style scores.

## 5.2 ADDRESSING L1 & L2 IMPROVES DIAGNOSTIC REASONING ON REAL-WORLD CASES

|  | Claude3-Opus | GPT-4o-mini | Huatuo-7B | Huatuo-34B | SemiHVision | SemiHVision-AN |
|---|---|---|---|---|---|---|
| **Accuracy** | 58.4 | 46.2 | 34.5 | 44.7 | 41.2 | **58.5** |
| **UMLS Factuality** | 0.18 | 0.16 | 0.13 | 0.16 | 0.11 | **0.23** |
| **GPT-4 Overall** | $1.17 \pm 0.04$ | $0.91 \pm 0.06$ | $1.08 \pm 0.03$ | $1.13 \pm 0.05$ | $0.78 \pm 0.04$ | **1.29**$\pm 0.02$ |
| **GPT-4 Key-Points** | 1.27 | 0.99 | 1.11 | 1.01 | 0.82 | **1.28** |
| **GPT-4 Inference** | **1.56** | 1.13 | 1.06 | 1.06 | 0.63 | 1.32 |
| **GPT-4 Evidence** | 0.67 | 0.60 | 1.08 | **1.31** | 0.89 | 1.27 |

Table 2: UMLS-F and GPT-4 score on JAMA Clinical Challenge across 6 different models :Claude3-Opus, GPT-4o-mini, Huatuo-GPT-Vision 7B, Huatuo-GPT-Vision 34B, SemiHVision, SemiHVision-AN. We also change Deepseek model to evaluate them to eliminate the bias as shown in Table 7

While public medical MLLMs often look strong on traditional benchmarks—occasionally even surpassing advanced general models like Claude3-Opus—a critical question remains: *Do medical MLLMs actually outperform general MLLMs on clinical tasks?* To answer this, we evaluate six models—Claude3-Opus, GPT-4o-mini, Huatuo-7B, Huatuo-34B, SemiHVision, SemiHVision-AN—on the JAMA Clinical Challenge using our evaluation pipeline (Table 2). We report both accuracy (standard close-ended QA) and diagnostic reasoning via the automatic scoring pipeline in Sec. 4.4, decomposed into Key Points, Inference, and Evidence. Despite strong traditional-benchmark results (e.g., SemiHVision-AN accuracy **58.5%**), models struggle on JAMA: Huatuo-34B excels on Evidence (**1.31**, higher than Claude's **0.67**) yet shows weaker Inference (**1.06**); in contrast, the general models Claude3-Opus and GPT-4o-mini achieve Inference **1.56** and **1.13**, respectively. These

findings indicate that larger medical-specific models can memorize domain facts without translating them into superior diagnostic reasoning—i.e., medical MLLMs do not necessarily outperform general MLLMs on clinical tasks requiring inference.

**Human study and reliability.** To corroborate the automatic pipeline, three medical professionals reviewed a 100-question sample and expressed preferences between rationales from Claude3-Opus and SemiHVision given the gold rationale. Results align with automation:SemiHVision-AN attains a **0.57** win rate over Claude3-Opus, supporting the reliability of our automatic evaluation.

**Training for robust diagnostic capability (annealing).** Addressing *How can we train a medical MLLM with robust diagnostic capabilities?*, we instruction-tune SemiHVision. Initially, it can answer medical QA but scores lower across metrics, particularly Inference (**0.63**), due to the absence of human-annotated diagnostic supervision. After applying annealing—pretraining on GPT-4o synthetic data then fine-tuning on human-annotated diagnostic data—the enhanced SemiHVision-AN achieves the top GPT-4 Overall score **1.29** and competitive accuracy **58.5%**. This demonstrates that integrating high-quality, human-annotated diagnostic data substantially improves diagnostic reasoning and can surpass models trained solely on synthetic or unannotated data (e.g., PubMedVision).

**Why fixing L2 helps—and why traditional metrics can mislead.** Traditional VQA sets are dominated by *knowledge* items (SLAKE **78.1%**, VQA-RAD **76.4%**, PathVQA **69.2%**), so models can score well by recalling single-view facts or template associations; JAMA contains far fewer knowledge items (**44.9%**) and instead requires *inference* from *multiple* views/modalities (Fig. 5). This mismatch explains why surface accuracy on VQA can overstate clinical readiness: those tests rarely force cross-view corroboration, ROI grounding, or contradiction checks. Addressing **L2** changes the mechanics of reasoning: (i) triangulation—the model must align findings across AP/LAT, sequences/slices, and modalities to confirm or refute a hypothesis (e.g., pneumonia vs. atelectasis); (ii) disambiguation—look-alikes are separated by view-dependent cues (projection, windowing, phase) and linked to clinical signals (labs, history); (iii) attribution

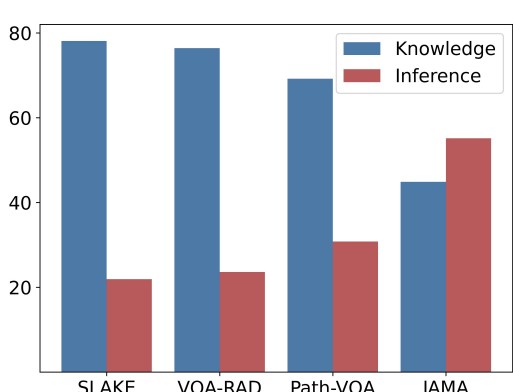

Figure 5: This figure illustrates the proportion of questions assessing knowledge and inference in the Slake, VQA-RAD, Path-VQA, and JAMA Clincial Challenge datasets.

and counterfactuals—claims must cite specific ROIs/views and remain consistent if a view is removed or replaced, reducing shortcut heuristics; and (iv) temporal/study coherence—evidence must agree across related images from the same case. Once these constraints are learned, JAMA's rubric dimensions improve because they directly reward multi-evidence synthesis: Inference rises from **0.63** to **1.32**, Overall from $0.78 \pm 0.04$ to $1.29 \pm 0.02$, and UMLS-F1 from **0.11** to **0.23**. As a judge-bias check, re-scoring with DeepSeek yields consistent conclusions (Table 7).

## 6 CONCLUSION

In summary, we diagnose two root causes of current medical MLLM underperformance—**L1** the absence of end-to-end diagnostic workflow supervision and **L2** the lack of multi-view/multi-modal evidence alignment—and address both with SemiHVision, a case-centric, study-level, multi-stage instruction corpus. Trained first for workflow competence and then annealed on expert, study-level data to learn cross-view synthesis, SemiHVision-AN attains SOTA on traditional medical VQA while also delivering large gains on the JAMA Clinical Challenge, where evidence-linked reasoning is required. Our evaluation pipeline—combining accuracy, UMLS-F1, and a blinded rubric on Key Points, Inference, Evidence—confirms that improvements are not merely stylistic: models trained on SemiHVision generalize better across standard tasks after fixing L1 and, after fixing L2, more reliably integrate corroborating findings across views/ROIs and clinical signals to support real clinical diagnosis. SemiHVision thus provides both the data recipe and training protocol needed to convert caption-style knowledge into clinically grounded diagnostic reasoning.

## 7 LIMITATIONS AND ETHICAL CONSIDERATIONS

Despite the promising results demonstrated by SemiHVision-AN, several limitations warrant consideration. Firstly, the coverage of anatomical regions in our dataset is limited due to the scarcity of high-quality, human-annotated medical data. While we have incorporated multiple imaging modalities such as X-ray, CT, and MRI, the representation across different body parts remains uneven. This imbalance may affect the generalizability of our model in diverse clinical scenarios, potentially limiting its performance on underrepresented regions. Additionally, the model size is constrained to 8 billion parameters, which, while efficient for training and deployment, may restrict the ability to handle more complex reasoning tasks that require deeper understanding and broader context. Exploring larger model architectures could enhance diagnostic performance in future work.

Moreover, the broader societal impacts of deploying SemiHVision-AN necessitate careful consideration. Automated medical systems hold significant potential for improving healthcare efficiency and accuracy but could also influence the roles of medical professionals and patient care practices. It is crucial to approach the implementation of such technological solutions with caution, ensuring they serve as a complement rather than a replacement to the expertise of healthcare professionals. Balancing technological advancement with ethical considerations is essential to maximize benefits while mitigating potential risks in clinical practice.

## 8 REPRODUCIBILITY STATEMENT

We have taken several steps to ensure the reproducibility of our work. All models and algorithms are described in detail in the main text (Sections 3), with theoretical formulations of the motivation provided in section 3. The description of datasets and preprocessing steps is given in Section 4. Hyperparameters and training configurations are reported in Section 4.

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

# A  APPENDIX

## A.1  LLM USAGE

In accordance with the ICLR 2026 policies on LLM usage, we disclose how LLMs were used in this work. LLMs were employed to assist with grammar polishing, wording improvements, and drafting text during paper preparation. All technical content, proofs, experiments, and analyses were conceived, implemented, and validated by the authors. Authors remain fully responsible for the correctness of the claims and results.

No LLMs were used to generate research ideas or produce results. No confidential information was shared with LLMs, and no prompt injections or other inappropriate uses were involved.

This disclosure aligns with the ICLR Code of Ethics: contributions of tools are acknowledged, while accountability and verification rest entirely with the human authors.

## A.2  TEMPLATE PROMPT

**Generate Instruction Data** In constructing our instruction dataset, we utilize both closed-ended and open-ended question formats. For closed-ended data, such as PMC-VQA, Amboss VQA, JAMA train VQA, Slake train VQA, VQA-RAD train, and Path VQA, we generate answer options only. For open-ended tasks, particularly from JAMA datasets, we also require the model to provide reasoning along with the answers. Additionally, GPT-4o is employed to generate question-answer pairs (QAPs) based on the images and their corresponding augmented captions, with each caption paired with 3 to 10 QAPs depending on its length and complexity. The questions generated are carefully designed to be directly related to the images, ensuring that answers can either be explicitly found or inferred from the caption content. The template prompt deatils are shown in Table 4. This approach minimizes dataset's hallucinations by grounding GPT-4o's output in the information provided in the captions and image data. Furthermore, we utilize a multigranular informtaion, such as specific ROI, and the broader medical context that connects local and global abnormalities to improve model's fine grained ability. By following this structured methodology, we ensure the generation of high-quality, clinically relevant instruction data that improves the accuracy and interpretability of the models.

**Evaluation Pipeline Prompt:** When evaluating close QA, we only need to calculate accuracy. However, many open QA tasks, such as diagnostic reasoning questions in the JAMA Clinical Challenge, present additional challenges. Although several methods exist for measuring textual similarity, such as F1 or ROUGE, both approaches have significant limitations in the medical domain. Therefore, we propose a very strict evaluation pipeine by using two evaluation metrics: the USMLE-Factuality score and the GPT-4o score. For the GPT-4o score, directly allowing GPT-4o to grade the

answers is often ineffective, as GPT-4o tends to favor answers that align with its preferred linguistic style, which may not match our intended criteria. Thus, we introduce a scoring framework to evaluate model's fine grained diagnostic ability based on three aspects: **Key Points**, **Inference**, and **Evidence** which is designed by doctors(The details are shown in Appendix A.2):

- **Key Points** assess whether the model's answer includes the critical elements present in the ground truth.

- **Inference** evaluates whether the diagnostic reasoning in the model's answer is correct, follows the same steps as the ground truth, and whether any key steps are omitted.

- **Evidence** examines whether the model's answer provides the crucial evidence to support its conclusions or diagnostic reasoning.

Finally, an average score will be calculated to represent the overall quality of the answer. To further reduce the influence of linguistic style on GPT-4's scoring, we propose revising all model-generated answers through GPT-4, ensuring that all outputs align with GPT-4's own style distribution. During this revision, GPT-4 will only see the model's answer, without access to any other information.

When scoring, GPT-4 will generate its own summaries of **Key Points**, **Inference**, and **Evidence** based on the ground truth. When assigning scores to these aspects, GPT-4 will no longer see the original answer but will only reference its summarized **Key Points**, **Inference**, and **Evidence**. For further details, please refer to Table 5, 6.

| Model | VQA-RAD (Finetuned) | SLAKE (Finetuned) | PathVQA (Finetuned) | PMC-VQA (Finetuned) | Avg. |
|---|---|---|---|---|---|
| **Fine-tuning on the training set.** | | | | | |
| LLAVA-v1.5-LLAMA3-8B | 63.3 | 68.9 | 85.2 | 50.3 | 66.9 |
| LLAVA_Med-8B | 66.3 | 69.5 | 90.7 | 52.7 | 69.8 |
| HuatuoGPTVision-8B | 68.9 | 84.1 | 93.0 | 57.3 | 75.8 |
| SemiHVision | 88.3 | 91.1 | 92.7 | 88.6 | 90.2 |

Table 3: Finetuning results on VQA-RAD, SLAKE, PathVQA, and PMC-VQA datasets.

## A.3 INSTRUCTION TUNNING

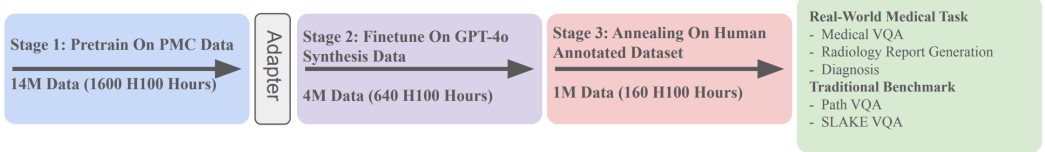

Figure 6: We apply three stages to train SemiHVision.

We employed an annealing strategy in training SemiHVision-AN to enhance its diagnostic capabilities. Empirically, annealing on small amounts of high-quality, human-annotated data significantly boosts performance on key benchmarks. Similar to Llama3, we performed annealing with a data mix that prioritizes high-quality data in select domains, excluding any training sets from commonly used benchmarks. This approach allowed us to assess the true few-shot learning capabilities and out-of-domain generalization of SemiHVision-AN.

We evaluated the efficacy of annealing on the JAMA Clinical Challenge and other diagnostic reasoning benchmarks. The annealing process substantially improved the performance of the pre-trained SemiHVision-8B model, demonstrating enhanced reasoning abilities and clinical applicability. These improvements suggest that, even with a model size constrained to 8 billion parameters, strategic annealing with high-quality data can compensate for limitations in model scale, enabling the model to handle complex reasoning tasks requiring deeper understanding. The whole training phase is shown in figure 6.

Table 4: Generate Instruction Data Prompt Example Template.

| | |
|---|---|
| **System Prompt** | Analyze the provided MRI image and generate a detailed and professional medical report that describes only the abnormalities, significant features, or relevant observations directly seen in the image. Use precise medical terminology and maintain a formal tone. Do not include any introductory phrases, such as "The provided image reveals," or any concluding remarks. Here are some relevant medical guidelines and Clinical cases for you to generate. |
| **Medical Guideline** | Angioplasty (PTBA) of the hepatic vein is a safe and effective treatment for Budd-Chiari syndrome (BCS) caused by hepatic venous outflow obstruction. This study, conducted between September 1996 and October 2008, included 101 patients (52 males, 49 females) with a mean age of 31.3 years, all presenting with symptomatic portal hypertension. Of these, 92 patients underwent successful PTBA, targeting the right, left, or accessory hepatic veins, with a technical success rate of 91%. PTBA significantly reduced hepatic venous pressure... |
| **Instruction Prompt** | Your second task is to generate 1-2 valuable questions and their corresponding answers that are relevant to the image's content and it would be better that the answers could be explicitly found within the discussion. |
| **Clinical Case** | 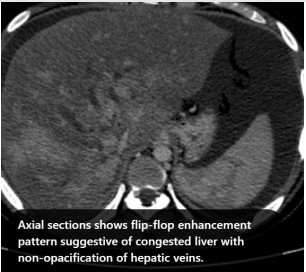 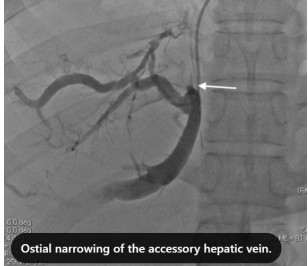 **Image Findings:** The patient underwent contrast-enhanced computed tomography which showed features of a congested liver with flip-flop pattern of enhancement. Hepatic veins show hypoattenuation on delayed phase. An accessory hepatic vein is also noted in segment VI. A diagnosis of Budd Chiari syndrome (BCS) was made on the basis of the clinical and imaging features. The patient was referred to the interventional radiology team for an endovascular rescue. On conventional venogram, the diagnosis of BCS was confirmed as the hepatic veins were thrombosed. An accessory segment VI hepatic vein was noted draining into the IVC... |
| **Format Prompt** | Return the results in the following format: **Report:** report content **Question:**Question content **Answer:**Answer content. Don't generate any other information Here is the image and discussion: |
| **Title: Accessory right inferior hepatic vein** | 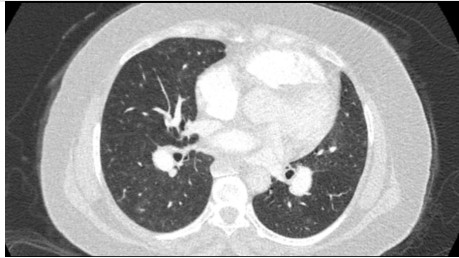 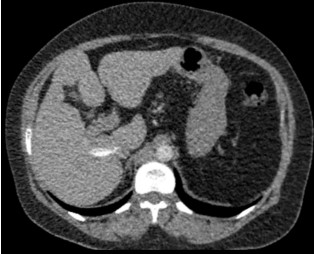 **Discussion:**Marked dilatation of the pulmonary trunk (6.7 cm) with the right (5.4 cm) and left (4 cm) main branches. Lung window shows mild bilateral diffuse faint groundglass centrilobular lung nodules that may reflect an underlying infection. Scans through the upper abdomen revealed average size cirrhotic liver and reflux of contrast into the IVC and hepatic veins with Incidental opacification of accessory right inferior hepatic vein... |

Table 5: Evaluation Pipeline Prompt Example Template.

| | |
|---|---|
| **Extract Key Points** | Based on the question and answer, summarize ten key points that you consider to be the most crucial from the standard answer. Return the response in the following format: {1.2.3....} Here is the question:{question} Here is the answer:{answer} Please do not provide any additional information. |
| **Key Points** | 1. Multifocal electroretinogram (ERG) showed reduced signal in the right eye throughout the macula, confirming the diagnosis of AZOOR.2. Acute zonal occult outer retinopathy (AZOOR) was first described by Gass in 1993... |
| **Extract Diagnostic Reasoning** | Based on the question and answer, please provide a detailed summary of the diagnostic reasoning from the standard answer. Return the response in the following format: {1.2.3....} Here is the question:{question} Here is the answer:{answer} Please do not provide any additional information. |
| **Diagnostic Reasoning** | 1. The patient is a 7-year-old boy with a slowly growing, asymptomatic lump on the left lower neck since birth.2. Physical examination showed a yellowish, hump-like mass with a hairy surface and cartilage-like consistency near the left sternocleidomastoid muscle... |
| **Extract Evidence** | Based on the question and answer, please provide a detailed evidence list which is proposed by correct answer. Return the response in the following format: {1.2.3....} Here is the question:{question} Here is the answer:{answer} Please do not provide any additional information. |
| **Evidence** | 1. Slowly growing, asymptomatic lump on left lower neck since birth.2. Physical examination revealed a yellowish, hump-like mass with hairy surface and cartilage-like consistency.3. Ultrasonography indicated a hypoechoic, avascular, bulging nodule with an anechoic tubular structure.4. MRI demonstrated a protuberant nodule with diffuse... |
| **Key Points Score** | Act as a USMLE evaluator, your role involves assessing and comparing a medical student's explanation to the provided target answer. Begin the assessment by carefully reviewing the provided target answer. Then, based on following specific criteria, determine the score for the student's answer. Please judge whether medical student's answer include these key points(or some other relevant points. But the amount of points must be complete). For example, ground truth have 10 key points, if student answer include one key he will get 0.5 point(if the answer include 5 points so should be 2.5). Medical student's answer: {answer} Key Points: {Key Point} Please only return a float number(from 0 to 5). You should check each point one by one(shouldn't judge based on language style such as fluence and so on. Only judge based on whether the student's answer include correct or relevant and complete key points). Don't generate any other information. |

## A.4 BASELINE & BENCHMARK

**Medical MLLMs**: We evaluated three medical multimodal large language models (MLLMs): Med-Flamingo Moor et al. (2023), RadFM Wu et al. (2023), LLaVA-Med-7B Li et al. (2024) and HuatuoGPTVision-34B Chen et al. (2024b).

**General MLLMs**: We assessed the latest models from the LLaVA series, including LLaVA-v1.6-7B, LLaVA-v1.6-13B, and LLaVA-v1.6-34B Liu et al. (2024a). Additionally, we compared these models with Yi-VL-34B Young et al. (2024) and Qwen-VL-Chat Bai et al. (2023). Additionally, we also evaluated several closed-source models: GPT-4-O-Mini and Claude3-Opus.

To evaluate the medical multimodal capabilities of the MLLMs, we employed two types of benchmarks:

**Medical VQA Benchmark**: We used the test sets from VQA-RAD Lau et al. (2018), SLAKE Liu et al. (2021), PathVQA He et al. (2020), and PMC-VQA Zhang et al. (2023) to assess the models' medical question-answering abilities. The experiment settings are the same as HuatuoGPT Vision.

**New Diagnosis Reason Benchmark Task**: To test the model's inference and medical knowledge capabilities, we will evaluate several medical multimodal models on the JAMA Clinical Challenge datasets. The JAMA Clinical Challenge dataset presents complex real-world cases from the Journal of the American Medical Association, challenging models with diagnostic and management tasks based

Table 6: Evaluation Pipeline Prompt Example Template.

| | |
|---|---|
| **Inference Score** | Act as a USMLE evaluator, your role involves assessing and comparing a medical student's explanation to the provided target answer. Begin the assessment by carefully reviewing the provided target answer. Then, based on following specific criteria, determine the score for the student's answer. Please judge whether medical student's answer's diagnostic reasoning is correct based on ground truth. For example, ground truth have 10 steps, if student answer include one correct step he will get 0.5 point(if student have other correct diagnostic reasoning path it should also be correct. But the amount of evidence must be complete. It means that each step is about 0.5 point if there are 10 steps). Medical student's answer: {answer} Ground Truth: {diagnostic reasoning} Please only return a float number(from 0 to 5). You should check each step one by one(shouldn't judge based on language style such as fluence and so on. Only judge based on whether student's diagnostic reason is correct or relevant). Don't generate any other information. |
| **Evidence Score** | Act as a USMLE evaluator, your role involves assessing and comparing a medical student's explanation to the provided target answer. Begin the assessment by carefully reviewing the provided target answer. Then, based on following specific criteria, determine the score for the student's answer. Please judge whether medical student's answer provide detail evidence such as ground truth. For example, ground truth have 10 evidence, if student answer include one evidence he will get 0.5 point(if student give other correct detail evidence, it is also correct. But the amount of evidence must be complete.) Medical student's answer: {answer} Detail Evidence: {evidence} Please only return a float number(from 0 to 5). You should check each evidence one by one(shouldn't judge based on language style such as fluence and so on. Only judge based on whether student propose correct and complete diagnostic evidence). Don't generate any other information. |

on clinical data and imaging. Together, these datasets provide rigorous benchmarks for assessing the diagnostic and decision-making performance of MLLMs in real-world clinical settings.

### A.5 FINE-TUNED RESULTS

To assess the impact of SemiHVision on downstream tasks, we applied fine-tuning using the benchmark training sets. As illustrated in Table 3, SemiHVision substantially enhances performance in downstream medical tasks, providing notable improvements across all four VQA tasks.

### A.6 LANGUAGE STYLE INFLUENCE

While our method still utilizes GPT-4o, it effectively eliminates the influence of language style. This is because our scoring is based primarily on whether key points are covered and whether there are any hallucinated key points. Each key point corresponds to a separate score, so variations in language style do not affect the outcome—language style won't cause the model to include more or fewer key points. It's true that switching to a different evaluation model may lead to slight differences in the extracted key points, which could influence the absolute score. However, keep in mind that these key points are derived from the ground-truth answer, and LLMs generally perform very well in summarization tasks. So while there may be changes(for example some model will summarize the most five key points but GPT4o will summarize 10 points), they do not affect the relative ranking of the scores. For fairness, we also evaluated the subset of data using DeepSeek as the scoring model. As shown in the Table 7, although the absolute values differ slightly, the relative scores remain consistent.

### A.7 FACTUALITY METRICS: UMLS-F1

To evaluate the factual accuracy of LLM outputs, we leverage the UMLS concept overlap metric. The Unified Medical Language System (UMLS) Bodenreider (2004) enhances biomedical interoperability

Table 7: Performance comparison across different models. Bold indicates best performance.

| Model | Claude3-Opus | GPT-4o-mini | Huatuo-7B | Huatuo-34B | SemiHVision | SemiHVision-AN |
|---|---|---|---|---|---|---|
| Accuracy | 58.4 | 46.2 | 34.5 | 44.7 | 41.2 | **58.5** |
| UMLS Factuality | 0.18 | 0.16 | 0.13 | 0.16 | 0.11 | **0.23** |
| GPT-4 Overall | 1.17 | 0.91 | 1.08 | 1.13 | 0.78 | **1.29** |
| DeepSeek Overall | 2.31 | 1.95 | 2.06 | 2.24 | 1.86 | **2.55** |

by unifying a comprehensive collection of biomedical terminologies, classification systems, and coding standards. By reconciling semantic variances and representational disparities across different biomedical repositories, UMLS facilitates standardized understanding.

We employ the Scispacy library[2] to identify and align medical named entities in texts with their corresponding UMLS concepts. Scispacy excels in entity recognition, enabling accurate association of named entities in LLM outputs with relevant UMLS concepts, a critical capability for assessing factual accuracy.

Our analytical process utilizes precision and recall metrics. Precision measures the proportion of shared concepts between the LLM output and the ground truth, indicating factual correctness. Recall assesses how well the LLM output covers the concepts present in the ground truth, reflecting the relevance of the information. Formally, given the concept sets from the ground truth ($C_{\text{ref}}$) and the LLM output ($C_{\text{gen}}$), precision and recall are calculated as:

$$\text{Precision} = \frac{|C_{\text{ref}} \cap C_{\text{gen}}|}{|C_{\text{gen}}|}, \tag{1}$$

$$\text{Recall} = \frac{|C_{\text{ref}} \cap C_{\text{gen}}|}{|C_{\text{ref}}|}. \tag{2}$$

The F1 score, derived from these precision and recall values, provides a balanced measure of the LLM output's accuracy and relevance.

## A.8 DATA SOURCE

The fine-tuning datasets include DeepLesion, MIMIC-CXR-JPG, PadChest, Quilt, LLD-MMRI, and MAMA-MIA, along with benchmark training QA datasets such as VQA-RAD, Path VQA, PMC VQA, and Slake VQA, covering multiple modalities like CT, MRI, X-ray and so on. Additionally, we expanded the dataset with data from Eurorad and Radiopaedia to include more diverse modalities as shown in table 8. Additionally, to enable the model to support multiple languages, such as Chinese, we randomly selected 300k datasets and translated them into Chinese for training.

## A.9 HUMAN EVALUATION AND CASE STUDY

**Case Study for Evaluation** We selected a case from the JAMA Clinical Challenge to evaluate the diagnostic reasoning capabilities of different models, as shown in Table 11[3]. In the case we apply three different colors: red, blue, brown to ask GPT-4O to annotated key points, inference points and evidence points. Our analysis revealed that Claude3-Opus performed accurate inference but lacked detailed evidential support. SemiHVision was able to generate diagnostic reasoning with comprehensive evidence, incorporating most of the important key points. In contrast, HuatuoGPTVision-34B and HuatuoGPTVision-7B failed to capture the essential key points and were unable to effectively utilize medical knowledge for detailed inference, despite having access to extensive medical information that could provide evidence.

**Human Annotated Sample Training Data** We sampled a case from EURORAD[4]. For EURORAD Dataset, there are serveral sections: Image Caption, Clinical History, Image Findings and Discussion as shown in Table 12. The Image Caption provides a concise description of each image presented.

---

[2]Using the Scispacy *en_core_sci_lg* model

[3]The case is sourced from https://jamanetwork.com/journals/ jamaophthalmology/fullarticle/2681464.

[4]The case is sourced from https://www.eurorad.org/case/16705.

| Dataset | Data Size | Modality | ROI | Human Annotation | Slice ID |
|---|---|---|---|---|---|
| Deeplesion | 24,821 | CT | × | × | × |
| PadChest | 150,730 | CT | × | ✓ | - |
| Eurorad | 691,370 | CT,X-Ray,MRI...(Multi) | ✓ | ✓ | ✓ |
| MIMIC-CXR-JPG | 620,113 | X-Ray | × | ✓ | - |
| LLD | 30,390 | MRI | ✓ | × | ✓ |
| MAMA-MIA | 76,381 | MRI | ✓ | × | ✓ |
| PMC-VQA | 152,603 | CT,X-Ray,MRI...(Multi) | × | ✓ | - |
| Path-VQA | 19,654 | Pathology | × | ✓ | - |
| PMC-Instruct | 619,606 | CT,X-Ray,MRI...(Multi) | × | ✓ | - |
| Quilt | 1,017,416 | Histopathology | × | ✓ | - |
| Radiopaedia | 1,131,614 | CT,X-Ray,MRI...(Multi) | ✓ | ✓ | ✓ |
| SLAKE | 9,835 | CT,X-Ray,MRI | × | ✓ | - |
| VQA-RAD | 1,798 | X-Ray,MRI | × | ✓ | - |
| AMBOSS & JAMA | 45,820 | Multi & Only Text | ✓ | ✓ | - |
| Chinese Data | 300,000 | Multi | - | - | - |

Table 8: Data Source.

Table 9: Distribution of Articles Across JAMA Specialty Journals

| Journal | Count |
|---|---|
| JAMA Otolaryngology–Head & Neck Surgery | 513 |
| JAMA Ophthalmology | 466 |
| JAMA Dermatology | 368 |
| JAMA (General) | 328 |
| JN Learning | 299 |
| JAMA Surgery | 133 |
| JAMA Oncology | 105 |
| JAMA Cardiology | 92 |
| JAMA Neurology | 61 |
| JAMA Pediatrics | 60 |
| JAMA Psychiatry | 6 |

The Clinical History records the patient's medical background and presenting symptoms. In the Imaging Findings section, experts analyze the images to arrive at a diagnostic conclusion, combining observations from all available imaging modalities. The Discussion elaborates on the inference steps and presents the evidence supporting the diagnosis, along with relevant background information to aid in understanding how the conclusion was reached. We also present one sample for our SemiHVision dataset.

**Case Study for Multimodality Retriever** We did a case study to prove the important of multi-modality retriever in our pipeline as shown in Table 13. The inclusion of a retriever in the image description task introduces a marked improvement in the specificity and accuracy of the generated descriptions. Without the retriever, the model (GPT-4o) provides a generalized description of the image, identifying broad anatomical landmarks (heart, aorta, and vertebral column) and speculating on potential abnormalities, such as a mass or vascular anomaly. While the description is coherent, it lacks precision, as the model does not have access to clinical guidelines or related cases, resulting in a speculative rather than a diagnostic interpretation.

In contrast, when the retriever is introduced, the model is supplemented with relevant clinical guidelines and case data, significantly enhancing its diagnostic accuracy. For example, in the case with the retriever, GPT-4o correctly identifies the subaortic ventricular septal defect (VSD) and provides a detailed explanation of its location, dimensions (2.7 cm), and potential clinical implications, such as abnormal blood flow and symptoms like shortness of breath. The addition of retriever-assisted information allows the model to go beyond general observations and offer more specific, clinically relevant insights, directly aligning the image interpretation with known medical cases.

| Dataset | Caption Available | License |
|---|---|---|
| DeepLesion | Yes | CC BY 4.0 |
| PadChest | Yes | PADCHEST Dataset Research Use Agreement |
| Eurorad | Yes | Creative Commons Attribution 4.0 International License |
| MIMIC-CXR-JPG | No | PhysioNet Credentialed Health Data License 1.5.0 |
| LLD | Yes | LLD-MMRI Agreement |
| MAMA-MIA | Yes | CC BY-NC-SA 4.0 |
| PMC-VQA | Yes | CC BY-SA |
| PMC-Instruct | Yes | OpenRAIL |
| Quilt | Yes | - |
| Radiopaedia | No | Radiopaedia Agreement |
| JAMA Clinical Challenge | No | JAMA Agreement |
| LLaVA-Med | Yes | CC BY-NC 4.0 |

Table 10: Overview of caption availability and dataset licenses.

**Human Annotator Information** We worked with six annotators, all of whom are experts. By experts, we mean either individuals with an MD degree or radiologists with over 10 years of clinical experience. For the image classification task, the three annotators hold MD degrees or work on radiology more than 10 years. For the subsequent human evaluation tasks, such as the one conducted on the JMLR dataset, we engaged three senior radiologists who assessed the model outputs with reference to the ground truth. Each of these doctors has more than ten years of professional experience.

Table 11: Sample Case in JAMA Clinical Challenge.

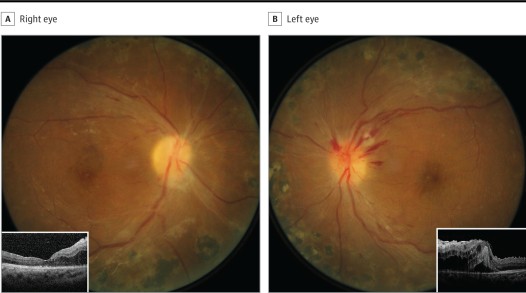

**Question:** A woman in her mid-20s presented with subacute bilateral vision loss that was worse in the left eye. Her medical history was remarkable for type 1 diabetes diagnosed at 16 years of age and proliferative diabetic retinopathy in both eyes that had been treated with panretinal photocoagulation 7 years earlier. She had undergone pars plana vitrectomy with endolaser to treat a tractional retinal detachment in her right eye 2 years before this presentation. She also had a history of hypertension and chronic kidney disease, and she was 15 weeks into pregnancy. Visual acuity was 20/50 OD and 20/100 OS. Intraocular pressure was normal bilaterally, and no relative afferent pupillary defect was detected. Findings of an anterior segment examination were normal. The patient was in no apparent distress and denied any headache, chest pain, or focal weakness. Ophthalmoscopic examination (Figure) revealed mild optic nerve head edema that was greater in the left eye than the right eye with associated nerve fiber layer hemorrhage in the left eye. Nerve fiber layer infarctions, dot and blot hemorrhages, and lesions caused by panretinal photocoagulation also were seen bilaterally. Optical coherence tomography showed macular edema that involved the center of the macula in both eyes (Figure, inset). A. Obtain a fluorescein angiogram B. Determine blood glucose level and perform glycated hemoglobin test C. Measure heart rate, respiratory rate, and blood pressure D. Perform immediate computed tomography of the head Answer with the option's letter from the given choices directly and give me the reason. Answer with the option's letter from the given choices directly and give me the reason

**Diagnostic Reason:** Malignant hypertension with papillopathy C. Measure heart rate, respiratory rate, and blood pressure The patient was found to have hypertension, with a blood pressure of 195/110 mm Hg. Heart and respiratory rates were normal. Measurement of the arterial blood pressure may be performed rapidly in the clinic with a sphygmomanometer and is essential to rule out malignant hypertension, which is a potentially life-threatening cause of vision loss. Although the differential diagnosis for bilateral optic nerve edema is broad, workup should always include assessment of blood pressure when appropriate, because a hypertensive emergency (also known as malignant hypertension) may cause substantial morbidity or mortality if not diagnosed and treated promptly. Findings may include macular star, macular edema, serous retinal detachment, intraretinal hemorrhage, and optic disc edema with or without associated hemorrhage.[1,2] Optic nerve head edema may occur with systolic blood pressures as low as 160 mm Hg, with the median onset occurring at 188 mm Hg.[3] The macular edema associated with hypertensive retinopathy may be distributed more nasally, as was seen in this patient.[4] This patient had mild optic nerve edema despite high systemic blood pressure and substantial macular edema. This less-pronounced optic nerve edema likely was attributable to optic nerve atrophy at baseline. Proliferative diabetic retinopathy and panretinal photocoagulation can be associated with optic atrophy, and atrophic optic nerves tend to become less edematous than healthy optic nerves.[5-7] Regarding the other choices above, a fluorescein angiogram (choice A) would be expected to show leakage from the optic nerve and macula, but such findings are already available from the optical coherence tomography, which showed intraretinal and subretinal fluid. Although assessment of serologic levels of glucose and glycated hemoglobin (choice B) is important in the management of diabetic retinopathy, results from such tests are not immediately available and have less bearing on the immediate management of the case (this patient's glycated hemoglobin level was 10.5% [to convert to a proportion of total hemoglobin, multiply by 0.01]).

Performing computed tomography (choice D) would be a reasonable later step, especially if the patient's blood pressure was found to be normal, to rule out an intracranial mass. In addition to hypertension, other causes of optic nerve head edema exist. Papilledema (when optic nerve head edema is secondary to increased intracranial pressure) is often bilateral and may be caused by intracranial mass lesions, meningitis, cerebral venous thrombosis, or intracranial hypertension, or it may be idiopathic. Bilateral optic nerve head edema also may be caused by diabetic papillopathy (which is a rare cause of optic disc edema, but which is bilateral in up to 50% of cases[8,9]) and toxic optic neuropathies. Optic disc drusen are frequently bilateral and may cause so-called pseudopapilledema.[10] Severe hypertension resulting in acute retinopathy and optic nerve head edema is considered a hypertensive emergency because of its association with end-organ damage, namely, damage to the retina and optic nerve. This case serves as a reminder that, although the differential diagnosis for optic nerve head edema is broad, immediately assessing the patient's blood pressure to evaluate for hypertensive emergency in the eye clinic may be critical, because this treatable condition could lead to substantial morbidity or mortality if diagnosis is delayed. The patient was admitted to the medicine service for management of hypertension. At an ophthalmology follow-up examination 3 months later, her blood pressure was 128/73 mm Hg and her visual acuity was 20/40 OD and 20/60 OS, close to her visual acuity before this acute event. On examination, the optic nerve head edema had resolved, and optical coherence tomography showed resolution of macular edema.

**SemiHVision:** Malignant hypertension with papillopathy C. Measure heart rate, respiratory rate, and blood pressure The key to the correct diagnosis is the presence of bilateral optic neuropathy in a pregnant woman. The patient's history of hypertension and chronic kidney disease, along with the findings of bilateral optic neuropathies, suggested a systemic etiology. A fluorescein angiogram (choice A) would not be the next step because the patient was asymptomatic and had no signs of retinal ischemia. A glycated hemoglobin test (choice B) would not be indicated because the patient had no signs of diabetic retinopathy. A computed tomography scan of the head (choice D) would not be appropriate because the patient had a normal examination and no signs of intracranial hemorrhage. Besides hypertension, other causes of optic nerve head edema exist. Papilledema, which is optic nerve head edema secondary to increased intracranial pressure, is often bilateral and may result from intracranial mass lesions, meningitis, cerebral venous thrombosis, intracranial hypertension, or it may be idiopathic. Bilateral optic nerve head edema can also be caused by diabetic papillopathy—a rare cause of optic disc edema that is bilateral in up to 50% of cases[1]—and toxic optic neuropathies. Optic disc drusen, which are frequently bilateral, may lead to so-called pseudopapilledema. Severe hypertension causing acute retinopathy and optic nerve head edema is considered a hypertensive emergency due to its association with end-organ damage, specifically affecting the retina and optic nerve. This case emphasizes that, although the differential diagnosis for optic nerve head edema is extensive, promptly assessing the patient's blood pressure in the eye clinic to evaluate for a hypertensive emergency is critical. Immediate evaluation is crucial because this treatable condition could lead to significant morbidity or mortality if diagnosis is delayed. The patient was admitted to the medical service for hypertension management. At an ophthalmology follow-up examination three months later, her blood pressure was 128/73 mm Hg, and her visual acuity improved to 20/40 in the right eye (OD) and 20/60 in the left eye (OS), nearing her visual acuity before this acute event. Upon examination, the optic nerve head edema had resolved, and optical coherence tomography showed a resolution of the macular edema.

**Claude3-Opus:** This case presents several concerning features that make measuring vital signs the most urgent next step: Bilateral optic nerve edema (worse in left eye) in a young pregnant woman with:

- Type 1 diabetes
- Hypertension
- Chronic kidney disease
- Active pregnancy (15 weeks)

The key concern here is POSTERIOR REVERSIBLE ENCEPHALOPATHY SYNDROME (PRES), which is:

- A neurological condition characterized by vasogenic edema
- Commonly associated with severe hypertension
- More common in pregnant women
- Can present with visual symptoms and optic nerve edema
- Can be life-threatening if not treated promptly

Why other options are less appropriate:

- A (Fluorescein angiogram): Contraindicated in pregnancy
- B (Blood glucose/HbA1c): While important for diabetic management, not the most urgent concern given the presentation
- D (Head CT): While eventually needed if PRES is suspected, vital signs should be checked first as they'll guide immediate management

The vital signs are crucial because:

- They will help determine if this is a hypertensive emergency requiring immediate intervention.
- They're non-invasive and provide immediate critical information

The findings will guide the urgency and direction of further workup

- If severely elevated blood pressure is found, immediate treatment could prevent serious complications for both mother and fetus
- This represents a potential medical emergency where checking vital signs is the most appropriate first step to guide immediate management decisions.

**Huatuo-GPT-Vision-34B:** The patient, who has a history of type 1 diabetes and is currently 15 weeks pregnant, presents with optic nerve head edema, macular edema, and hemorrhages in both eyes; therefore, it is crucial to assess her blood glucose levels and perform a glycated hemoglobin (HbA1c) test to evaluate and manage her glycemic control status, as this evaluation is essential to identify any acute changes or complications related to her pregnancy and existing diabetes, since poor glycemic control can lead to rapid progression of diabetic retinopathy, exacerbate ocular conditions, and pose significant risks to both maternal and fetal health—including the potential for congenital anomalies, preeclampsia, and worsening of chronic kidney disease—thus, timely assessment and adjustment of her diabetes management plan are vital to prevent further vision loss and ensure the well-being of both the mother and the developing fetus.

**Huatuo-GPT-Vision-7B:** B. Determine blood glucose level and perform glycated hemoglobin test Reasoning: The patient has a history of type 1 diabetes, which is known to cause diabetic retinopathy leading to vision loss as seen in the images provided (optic nerve head edema with hemorrhage). The presence of macular edema also suggests ongoing or poorly controlled diabetes affecting her eyes. A fluorescein angiogram would be useful for assessing vascular abnormalities but may not directly address the underlying issue related to poor glycemic control. Measuring heart rate, respiratory rate, and blood pressure are important general assessments but do not specifically relate to the ophthalmic findings presented here. Immediate computed tomography of the head might be considered if there was suspicion of other complications like intracranial pathology; however, it's less relevant given the current clinical context focused on eye examination results linked to diabetes management issues.

**GPT-o-mini:** Correct Answer: C. Measure heart rate, respiratory rate, and blood pressure. Reason: Measuring blood pressure is critical in this scenario to identify malignant hypertension, a potentially life-threatening condition that can lead to vision loss through optic nerve and retinal damage. Immediate assessment of blood pressure helps rule out hypertensive emergencies, allowing for timely intervention.

**Expert Comment:** SemiHVision and Claude3-Opus accurately recognized the urgency of measuring vital signs to assess for malignant hypertension in this patient, who presents with bilateral optic nerve edema and a history of hypertension and chronic kidney disease during pregnancy. Their reasoning reflects a proper understanding of the need for immediate intervention to prevent serious complications. On the other hand, Huatuo-GPT-Vision-34B and Huatuo-GPT-Vision-7B focused on assessing glycemic control by selecting to determine blood glucose levels and perform a glycated hemoglobin test. While managing diabetes is important, they failed to prioritize the immediate life-threatening condition suggested by the patient's symptoms, thus overlooking the critical need to rule out a hypertensive emergency. GPT-4o-mini could get the correct answer and some key points but lack lots of detail evidence to prove it.

Table 12: Human Annotated Sample Case.

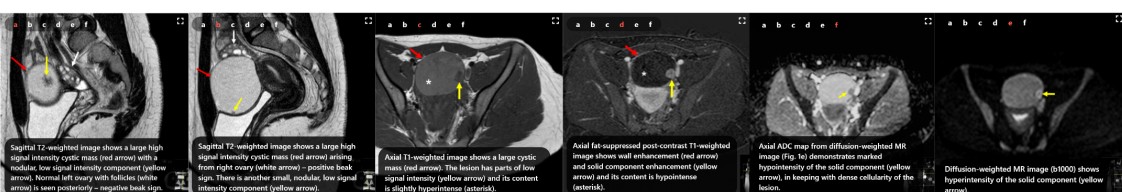

**Image Caption:** 1. Sagittal T2-weighted image shows a large high signal intensity cystic mass (red arrow) with a nodular, low signal intensity component (yellow arrow). Normal left ovary with follicles (white arrow) is seen posteriorly – negative beak sign. 2. Sagittal T2-weighted image shows a large high signal intensity cystic mass (red arrow) arising from right ovary (white arrow) – positive beak sign. There is another small, nodular, low signal intensity component (yellow arrow). 3. Axial T1-weighted image shows a large cystic mass (red arrow). The lesion has parts of low signal intensity (yellow arrow) and its content is slightly hyperintense (asterisk). 4. Axial fat-suppressed post-contrast T1-weighted image shows wall enhancement (red arrow) and solid component enhancement (yellow arrow) and its content is hypointense (asterisk). 5. Diffusion-weighted MR image (b1000) shows hyperintensity of the solid component (yellow arrow). 6. Axial ADC map from diffusion-weighted MR image (Fig. 1e) demonstrates marked hypointensity of the solid component (yellow arrow), in keeping with dense cellularity of the lesion.

**Clinical History:** A 21-year-old G0P0 woman with no medical history was referred to our institution for a sonographically detected cystic right adnexal mass. She has a history of pelvic discomfort without other complaints. Physical examination was normal. Laboratory findings were also normal except for an elevated CA 125 65.2 U/mL (normal <35.0).

**Image Findings:** MRI examination revealed a cystic tumour arising from the right ovary with 7.5 cm. On T2-weighted images, the signal intensity of the cyst content was high and two small nodular peripheral solid components were detected, adhering to its internal wall, with low signal (Fig. 1a, b). The normal left ovary was present with follicles (Fig. 1a). On pre-contrast T1-weighted images, the mass exhibited slightly high signal intensity (Fig. 1c). On contrast-enhanced fat-suppressed T1-weighted images, wall enhancement and solid component enhancement were detected (Fig. 1d). Finally, the ADC map (Fig. 1f) from diffusion-weighted image (Fig. 1e) demonstrates marked hypointensity of the solid component, in keeping with its dense cellularity. Surgical excision was proposed and accepted by the patient. The histopathological investigation revealed a typical ovarian serous borderline tumour.

**Discussion:** Borderline ovarian tumours are uncommon ovarian neoplasms, intermediate between benign and malignant types, corresponding to 5% of all epithelial ovarian tumours. [1, 2] Serous borderline tumour represents the most common type of borderline tumours arising in the ovary, and typically, it is confined to the adnexa and presents an indolent course. [3] However, up to 6.8% of these tumours can progress to low grade serous carcinoma. [3] Serous borderline tumours are divided into typical (90%) and borderline tumours with micro-papillary patterns (5%–10%). [4] These neoplasms usually present as bilateral adnexal masses with more proliferation of papillary projections than do benign cystadenomas, they are often seen in younger patients, and laboratory findings show the serum CA-125 level mildly elevated. [2, 3, 5, 6] The peak age of presentation is 45 years. [5] Small tumours usually do not cause symptoms and are often detected as an incidental finding on sonography. [7] Larger or more advanced neoplasms might cause pain or pelvic discomfort. The diagnosis of this type of tumour is based on histopathological examination. As they are staged using the same ovarian cancer staging of malignant ovarian neoplasms [5], MRI plays a crucial rule in this evaluation. There are no pathognomonic imaging features of borderline ovarian tumours and they frequently resemble a benign lesion. Nevertheless, some pathologic and MR imaging studies suggested that large papillary projections are highly suggestive of borderline or malignant neoplasms. [1, 8] The preferred treatment is bilateral oophorectomy. Fertility sparing procedures such as unilateral oophorectomy or cystectomy are related with a higher incidence of recurrence when compared with bilateral oophorectomy (10-20% vs. approximately 5% for radical surgery). [3, 4] Young age (<30 years) has also been found to be a risk factor for recurrent disease in patients treated conservatively. As in this case, patients should be instructed to complete child-bearing within a short time frame following initial diagnosis, due to the high risk of recurrence. [3] The most important feature in predicting an adverse outcome in serous borderline tumours is the presence of serous lesions involving the peritoneum (so-called invasive implants). [3, 4] Tumours without invasive implants, as in this case, have been considered to be nonaggressive tumours, associated with a good prognosis and high survival rate. Follow-up using ultrasound imaging is mandatory, with particular attention paid to the remaining ovary in conservatively treated young patients. [4] Written informed patient consent for publication has been obtained.

**SemiHVision(only a sample for one image):**
**User:**What type of imaging technique is shown in Fig. 1a of the provided image?
**GPT:**The image shows a T2-weighted MRI scan.
**User:**Where is the cystic tumour located according to the image?
**GPT:**The cystic tumour is arising from the right ovary
**User:**What is the size of the cystic tumour?
**GPT:**The cystic tumour is 7.5 cm in size.
**User:**What does the red arrow in the image indicate?
**GPT:**The red arrow indicates the normal left ovary with follicles.
**User:**What does the yellow arrow in the image point to?
**User:**The yellow arrow points to one of the small nodular peripheral solid components adhering to the internal wall of the cystic tumour.
**User:**What characteristic of the cyst content is shown by the high signal intensity in the T2-weighted image?
**User:**The high signal intensity indicates that the cyst content is fluid-filled.
**User:**What is shown by the asterisk (*) in the image?
**User:**The asterisk (*) indicates the cystic tumour arising from the right ovary.

Table 13: Human Annotated Sample Case.

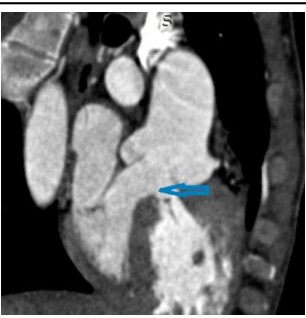

**Prompt(without retriever):** Generate the Image description based on the image.

**GPT-4o:** The image is a sagittal CT scan, likely of the thoracic or abdominal region. The scan shows various soft tissue structures, including major organs. The blue arrow points toward a specific structure, possibly a blood vessel or part of an organ. This could indicate an abnormality such as a mass, an aortic aneurysm, or an area of vessel narrowing or dissection. In the background, major anatomical landmarks like the heart, aorta, and vertebral column are visible, as well as some portions of the liver or spleen. The areas of high contrast indicate dense structures, such as bones, while softer tissues are seen in varying shades of gray, highlighting different densities. This scan could be used to evaluate conditions like vascular abnormalities, tumors, or other pathologies in the chest or upper abdominal region. Further clinical information would be necessary to confirm the exact diagnosis.

**Prompt(with retriever):** Generate the Image description based on the image. Here are some medical guidelines and Clinical cases for you.

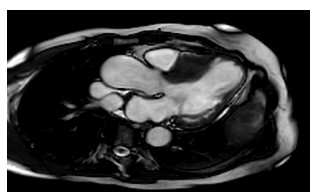
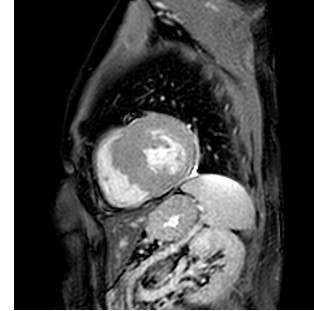

**Case Discussion:** A case of asymmetric hypertrophic cardiomyopathy with mild subaortic stenosis and partial systolic anterior motion of the mitral valve. Since the pressure gradient obtained by phase contrast imaging was rather low, echocardiographic reevaluation and/or follow-up was recommended. The case also demonstrates a typical pattern of myocardial fibrosis in the basal and midventricular septal segments at the anterior and posterior right ventricular insertion points.

**GPT-4o:** The image is a coronal oblique CT scan of the thoracic region, focused on the heart and its surrounding structures. The blue arrow indicates a subaortic ventricular septal defect (VSD), measuring approximately 2.7 cm. This congenital heart defect refers to an abnormal opening in the interventricular septum (the wall separating the left and right ventricles of the heart), located just below the aortic valve. The surrounding anatomy includes parts of the heart chambers, likely showing the left and right ventricles, as well as nearby vascular structures such as the aorta. The defect appears to allow abnormal communication between the ventricles, which could lead to altered blood flow dynamics, potentially causing symptoms such as shortness of breath, fatigue, or heart failure if left untreated.

