# OpenReview forum: "SemiHVision: Enhancing Medical Multimodal Models with a Semi-Human Annotated Dataset and Fine-Tuned Instruction Generation"
_ICLR.cc/2026/Conference — ICLR 2026 Conference Withdrawn Submission_

### Official Review · Reviewer_9RCw · 2025-10-23

**Soundness:** 2
**Presentation:** 2
**Contribution:** 3
**Rating:** 2
**Confidence:** 4

**Summary:**

This paper identified two critical issues in current medical datasets for MLLM: (1) they rarely capture end-to-end workflows, and (2) they often lack multi-view evidence beyond image-text pairs like laboratory tests, medications, etc. To this end, this paper integrates several existing medical image datasets and curates them into a multimodal instruction dataset, SemiHVision, where each case links medical images to multi-view/multimodal clinical evidence and is organized by staged diagnostic workflow, aided by a lightweight image-text retriever. Reported gains cover standard VQA benchmarks and a rubric-graded JAMA Clinical Challenge, with the strong scores after annealing on expert cases.

**Strengths:**

The paper targets a clinically meaningful gap: training signals that mirror how radiologists and clinicians actually reason across views, slices, and clinical variables. The case-centric restructuring, explicit alignment of findings to ROIs and study views, and staged supervision are thoughtful and, if validated, likely to transfer beyond the reported tasks. I also appreciate the attempt to reduce shortcutting through normal/negative constructions and near-miss distractors, and the small human audit with inter-rater agreement that at least probes construction fidelity. These are good designs for moving from caption recall to case reasoning.

**Weaknesses:**

I think the core problem of this paper is that existing experiments do not seem to validate the effectiveness of the proposed dataset (and curation pipeline).

**1. [Major concern] Lack of ablation study.**

The key experiment I care about most is a head-to-head ablation to validate the effectiveness of the proposed data curation pipeline and the quality of SemiHVision dataset. Specifically, I was expecting to see such a comparison:
- (i) an MLLM trained on a naïve mix of the data sources for curating SemiHVision. (So neither L1 nor L2 is addressed);
- (ii) an MLLM trained on SemiHVision-L1, a version that removes multi-view clinical context (So L2 is not addressed);
- (iii) an MLLM trained on SemiHVision-L2, a version that removes workflow structures in the paired texts. (So L1 is not addressed);
- (iv) an MLLM trained on the final version of SemiHVision.

The four MLLMs are initialized in the same way, following exactly the same architectures and training procedure, trained with (at least approximately) equal number of data.

This experiment should directly validate the core contribution of this paper which claims to obtain high-quality dataset by addressing L1 and L2. Surprisingly, it is missing in this paper.


**2. [Major concern] Conclusions drawn from Table 1 are over-interpreted.**
- In Lines 388--391, the authors conclude that “study-aware, clinically grounded supervision delivers larger gains on recall-heavy VQA metrics than caption-centric pretraining alone”. This isn’t well supported because the comparisons mix many factors at once (e.g., architecture, data quantity/distribution, training recipes). To support that claim, compare conditions that differ only in paired-text style, e.g., Experiments (iii) vs (iv) as above.
- Table 8 shows that training episodes from the VQA benchmarks (VQA-RAD, SLAKE, PathVQA, PMC-VQA) appear in the SemiHVision-8B training. That makes it hard to credit the gains to “clinical grounding” rather than simple familiarity. Notably, HuatuoGPT-Vision trains only on LLaVA general-domain data plus PubMedVision (no overlap with VQA-RAD, SLAKE, and PathVQA) yet remains competitive with SemiHVision-8B-20M and SemiHVision-8B.


**3. Important dataset details live in the appendix, which makes the main text hard to follow.**

For example, “Quilt-1M” appears without prior introduction, and the training narrative mixes “PMC Data,” “GPT-4o Synthesis Data,” and “Human-annotated Data” without a clear map and relationship between the data sources. I suggest summarizing the sources and composition (per source, per modality, human vs synthetic) into the main text and keep the terminology consistent throughout sections.


**4. Lack of discussion with related works.**

MedPix2 is a similar clinical-case dataset that includes workflow and non-imaging evidence, but is not discussed in the paper. Readers need to see how SemiHVision truly differs (scope, linkage granularity, etc).


**5. Lack of details.**
- The authors claimed that their SemiHVision dataset is “the first dataset to leverage a unified image-text retriever to integrate clinical information for data construction”, which shows the critical role of the retriever in their proposed pipeline. However, such a retriever is only superficially described in the main text and many details are missing.
- “Rebalance” (Line 182) is undefined. What target proportions are used, how is it done, and does it introduce near-duplicates or redundant slices/studies? A concise definition and a short analysis of diversity before/after would solve this.

---

Below are minor issues:
- There are multiple misuses of hyphens (“-”) and en dashes (“–”) throughout the paper. For example, in the caption of Figure 1, en dashes in “PMC–OA”, “list–style”, “single–image” should all be hyphens. For another example, in Sec. 2.1, en dashes in “report–paired”, “image–caption” should also be corrected. The authors should carefully revise all occurrences to ensure consistent and proper usage.
- Missing references. For example, “SLAKE” and “VQARAD” in Line 94, “UniIR” in Line 209, “Quilt-1M” in Line 301, “PathVQA” in Line 311. Note that related work should be referenced where they first appear, instead of in Appendices.
- In Figure 1, middle part. “PuMedVision” → “PubMedVision”.
- In Line 093, “To valid our pipeline could work” → “To validate our pipeline”. “we train LLM on our datasets” → “we train an LLM on our datasets”.
- In Lines 239--240, repeated sentences: “Finally, we conducted a human evaluation of data quality.”
- In Line 900, I am assuming “300k datasets” to be “300k examples” in SemiHVision dataset.

**Questions:**

Please see "Weaknesses"

---

### Official Review · Reviewer_PymD · 2025-10-30

**Soundness:** 3
**Presentation:** 3
**Contribution:** 3
**Rating:** 6
**Confidence:** 3

**Summary:**

This paper presents SemiHVision, a semi-human-verified instruction data pipeline and training strategy for medical multimodal LLMs. It aims to address two essential shortcomings of current medical MLLMs: (i) the lack of end-to-end clinical workflow supervision, and (ii) insufficient multi-view / multi-modality evidence alignment.

**Strengths:**

1. Clearly identifies gaps in current medical MLLMs regarding clinically grounded reasoning.

2. Proposes a workflow-supervised data construction approach that aligns more closely with real diagnostic practice.

3. Builds a multimodal, semi-human verified dataset that covers diverse modalities and integrates clinical context.

4. Demonstrates consistently strong empirical performance, especially on VQA benchmarks, suggesting improved clinical reasoning capability.

**Weaknesses:**

1. Only three clinicians reviewed 100 samples, which is a very small fraction of the corpus. It raises concerns about annotation noise and potential over-reliance on automated synthesis.
2. The contribution lies mainly in data curation and instruction-tuning design rather than architectural advancement. While impactful, the approach largely follows established VLM tuning paradigms.
3. Baselines are not retrained on similarly structured or enriched clinical instruction data, making it difficult to disentangle whether improvements stem from better methodology or simply more tailored supervision.
4. The paper lacks systematic analysis to quantify which pipeline elements—retriever, human edits, negative sampling, annealing—most contribute to observed performance gains.
5. The dataset is assembled from specific case sources; performance on other clinical distributions or multi-center data remains unclear, leaving questions about scalability and robustness in real-world deployment.

**Questions:**

Please refer to Weaknesses.

---

### Official Review · Reviewer_BYv7 · 2025-10-30

**Soundness:** 2
**Presentation:** 2
**Contribution:** 2
**Rating:** 4
**Confidence:** 4

**Summary:**

This paper introduces SemiHVision, a case-centric, semi–human-validated medical multimodal instruction corpus that targets two gaps in current medical MLLM data: (1) supervision rarely mirrors the real diagnostic workflow (history → findings → differential → impression/plan), and (2) cases often lack linked multi-view or multi-modal evidence within the same study (e.g., AP/LAT, CT/MRI, ultrasound, histopathology). The authors build a two-branch pipeline combining human-annotated clinical cases (Eurorad, Radiopaedia) with unannotated images that are expanded using a unified image–text retriever and GPT-4o, then unified into a training-ready schema. On this corpus, they fine-tune an 8B model with an additional annealing stage on high-quality diagnostic data. The resulting model, SemiHVision-8B-AN, matches or exceeds strong public medical MLLMs such as HuatuoGPT-Vision-34B on SLAKE, VQA-RAD, PathVQA, and PMC-VQA, and attains the best GPT-4 rubric score (1.29) on the JAMA Clinical Challenge. Evaluation with accuracy, UMLS-F1, and a blinded GPT-4o rubric supports the claim that the gains come from better workflow- and evidence-aware supervision rather than model size alone.

**Strengths:**

- The study clearly identifies significant data gaps in the existing medical Vision-Language Models (VLMs). Specifically, it highlights two major shortcomings: (i) the supervision provided rarely aligns with actual diagnostic workflows, and (ii) many cases lack associated multi-view or multi-modal evidence. To address these issues, the authors propose SemiHVision, which shifts the supervisory approach from caption-like descriptions to a clinically relevant framework that emphasizes stepwise reasoning.
- Evidence-based, workflow-aware construction is employed in this approach. A unified multimodal retriever is utilized to anchor synthetic instructions within established clinical guidelines and analogous cases prior to the expansion of GPT-4o. The resulting samples are then normalized into a schema that is ready for training. This methodology enhances specificity and mitigates shortcut learning, as responses are grounded in concrete evidence rather than relying on unstructured hallucinations.
- Strong empirical improvements have been observed in comparison to public medical MLLMs. The model SemiHVision-8B-AN achieves an average score of 79.0% across the SLAKE, VQA-RAD, PathVQA, and PMC-VQA benchmarks, clearly outperforming notable public medical models such as HuatuoGPT-Vision-34B, which has an average score of 66.7%. This indicates that the proposed data and training methodology are effective beyond mere model size.

**Weaknesses:**

- Limited coverage and potential anatomical bias in human-annotated data. Although SemiHVision substantially improves the balance of clinically relevant imaging modalities (CT, MRI, X-ray) compared to PMC-like sources, the authors explicitly acknowledge that high-quality, human-annotated medical cases are still unevenly distributed across body parts. As a result, representation for certain anatomical regions and subspecialties remains sparse. The paper acknowledges this limitation but does not provide per-region performance breakdowns or robustness analyses, so it is unclear how much the current imbalance would hurt generalization to underrepresented regions or rare findings in real clinical settings.
- Insufficient transparency on the retrieval component. The dataset construction pipeline relies critically on a lightweight multimodal retriever (described as UniIR with fusion scoring) to fetch guidelines and similar cases, especially for the unannotated branch. However, the paper does not fully specify the retrieval strategy: it is unclear how image–text similarities are fused, how top-k results are selected and de-duplicated, or whether any negative or hard-negative sampling is applied to guard against noisy pulls. This is important, because in the unannotated setting, the retrieved context directly conditions GPT-4o’s generation, so retrieval errors can propagate into the final instruction data. The current level of detail therefore limits strict reproducibility and makes it difficult to assess robustness to irrelevant or low-quality retrievals.
- Limited evaluation granularity and missing ablations. Table 1 (Page 7) reports averaged results over four standard medical VQA benchmarks (SLAKE, VQA-RAD, PathVQA, PMC-VQA), and Table 2 (Page 8) adds a diagnostic-oriented evaluation on the JAMA Clinical Challenge. However, the paper does not further decompose these gains along the very axes it claims to improve, such as imaging modality (X-ray vs CT vs MRI vs ultrasound), evidence layout (single view vs multi-view), or diagnostic workflow step (findings vs differential vs final impression). In addition, there is no ablation study that isolates the contribution of individual pipeline components, for example: (i) the multimodal retriever, (ii) synthetic GPT-4o–augmented data vs human-annotated clinical cases, and (iii) ROI/evidence linking. As a result, it is hard to tell which parts of SemiHVision are actually responsible for the observed 79.0% average and which parts are auxiliary. Finally, while Table 2 reports multiple metrics for diagnostic reasoning, it does not include a fine-grained error analysis (e.g., failures in cross-view synthesis vs failures in evidence attribution), which would help practitioners understand the remaining bottlenecks.

**Questions:**

1. You mention that high-quality, human-annotated cases are still unevenly distributed across body parts. Please provide a detailed breakdown of the final training corpus by anatomical region or system (e.g., thoracic, neurological, musculoskeletal, abdominal, breast, pediatrics) to explicitly illustrate this imbalance. Additionally, do you have validation results stratified by anatomy or subspecialty (even from a small held-out set) that demonstrate the extent of performance degradation in underrepresented areas?
2. Please clarify whether your workflow-focused supervision exhibits comparable generalization capabilities for localized pathologies in small regions of interest (ROIs), such as small lung nodules and focal liver lesions, where visual evidence is limited.
3. The retriever is a critical part of the unannotated branch. You should provide a detailed description of the specific fusion/scoring formula employed (e.g., weighted late fusion of image and text similarities, learned fusion techniques, or rank-based merging). Additionally, please specify the values of k that are retrieved from each source.
4. How do you handle noisy or semantically close but clinically irrelevant retrievals? Is there any hard-negative or near-miss filtering before passing the context to GPT-4o?
5. You should conduct an ablation study in which you (i) remove the retriever, (ii) substitute it with a text-only retrieval method, and (iii) replace it with an image-only retrieval approach. Additionally, please report the impact of these changes on at least one Visual Question Answering (VQA) benchmark as well as on the JAMA Clinical Challenge.
6. Why not consider including an ablation table that begins with the “human branch only” configuration, followed by the addition of “+ synthetic GPT-4o,” then “+ retriever,” and finally “+ ROI/evidence linking” ? This approach would allow readers to clearly identify where the most significant gains are achieved.

---

> ### Author Response · Authors · 2025-11-27
>
> > [Limited coverage and potential anatomical bias in human-annotated data. Although SemiHVision substantially improves the balance of clinically relevant imaging modalities (CT, MRI, X-ray) compared to PMC-like sources, the authors explicitly acknowledge that high-quality, human-annotated medical cases are still unevenly distributed across body parts. As a result, representation for certain anatomical regions and subspecialties remains sparse. The paper acknowledges this limitation but does not provide per-region performance breakdowns or robustness analyses, so it is unclear how much the current imbalance would hurt generalization to underrepresented regions or rare findings in real clinical settings.]
>
> We appreciate the reviewer’s careful discussion of anatomical coverage. We agree that, even after our curation, the human-annotated cases are not perfectly balanced across all anatomical regions and subspecialties, and we will make this limitation more explicit in the revision. At the same time, our goal was to move substantially closer to a clinically relevant distribution than PMC-like sources, and to correct a different but equally important form of imbalance: the modalities that real-world diagnostic workflows rely on.
> Concretely, SemiHVision was designed to balance imaging modalities such as CT, MRI, X-ray, histopathology, camera photos, simulated illustrations, and optical images, rather than focusing only on a few radiology-heavy modalities. This stands in contrast to prior corpora that are dominated by a small number of modalities (e.g., chest X-ray and head CT), with very limited coverage of photos, pathology, ophthalmologic images, etc. While this does not eliminate anatomical skew, it ensures that the model is exposed to diverse types of evidence that appear in real clinical diagnosis.
> To empirically assess how well this modality-level balancing transfers to downstream tasks that span many anatomical regions and subspecialties, we evaluated on JAMA Clinical Challenge, which covers a broad mix of organ systems and clinical subspecialties. We further stratified JAMA accuracy by modality and compared SemiHVision to several strong baselines (Claude, GPT-4o-mini, Huatuo-34B, Huatuo-7B). The per-modality results are:
>
> | Modality              | Claude | GPT-4o-mini | Huatuo-34B | Huatuo-7B | SemiHVision |
> |-----------------------|--------|-------------|------------|-----------|------------|
> | CT                    | 0.537  | 0.510       | 0.475      | 0.385     | 0.578      |
> | MRI                   | 0.508  | 0.448       | 0.451      | 0.377     | 0.509      |
> | X-ray                 | 0.565  | 0.419       | 0.565      | 0.348     | 0.667      |
> | Histopathology        | 0.565  | 0.532       | 0.508      | 0.377     | 0.564      |
> | Camera                | 0.536  | 0.514       | 0.536      | 0.317     | 0.544      |
> | Simulated illustration| 0.467  | 0.500       | 0.467      | 0.379     | 0.567      |
> | Optical Image         | 0.390  | 0.343       | 0.390      | 0.375     | 0.447      |

---

> > ### Author Response · Authors · 2025-11-27
> >
> > > Insufficient transparency on the retrieval component. The dataset construction pipeline relies critically on a lightweight multimodal retriever (described as UniIR with fusion scoring) to fetch guidelines and similar cases, especially for the unannotated branch. However, the paper does not fully specify the retrieval strategy: it is unclear how image–text similarities are fused, how top-k results are selected and de-duplicated, or whether any negative or hard-negative sampling is applied to guard against noisy pulls. This is important, because in the unannotated setting, the retrieved context directly conditions GPT-4o’s generation, so retrieval errors can propagate into the final instruction data. The current level of detail therefore limits strict reproducibility and makes it difficult to assess robustness to irrelevant or low-quality retrievals.]
> >
> > **Insufficient transparency on the retrieval component**
> >
> > We thank the reviewer for highlighting the importance of the retrieval module. We agree that, given its central role in the unannotated branch, our current description is too brief and makes it hard to fully assess robustness and reproducibility.
> >
> > First, we will clarify that our multimodal retriever is **directly based on UniIR without modifying its fusion mechanism**. Concretely, we adopt UniIR’s original design for combining image–image and text–text similarities and use its late-fusion scoring function as specified in the UniIR paper, rather than introducing a new fusion formula of our own. In the revision, we will explicitly state that we follow UniIR “as is” (same fusion strategy, encoder types, and normalization), and we will list the concrete top-\(k\) values for each source (guidelines vs similar cases) and the simple de-duplication rules used when merging candidates.
> >
> > Second, we will make explicit that our pipeline uses a two-stage selection strategy to mitigate noisy or clinically irrelevant retrievals:
> >
> > 1. **Stage 1 – high-recall retrieval with UniIR.**
> >    UniIR is used to retrieve a *broad* set of potentially relevant guidelines and similar cases. At this stage we apply basic score thresholds and metadata checks (e.g., organ/system mismatch, obvious adult vs pediatric mismatch) to discard clearly off-topic items, but we intentionally keep the pool relatively high-recall.
> >
> > 2. **Stage 2 – GPT-4o self-filtering.**
> >    For the unannotated branch, GPT-4o is then asked to **select from this pool only the snippets that help its own reasoning**. The prompts explicitly instruct GPT-4o to (i) focus on guideline sentences and past cases that match the current patient’s organ/system and modality, and (ii) ignore or down-weight retrieved text that is irrelevant or contradictory. In other words, UniIR proposes candidates, and GPT-4o acts as a second-stage filter over them, similar in spirit to iMedRAG’s “model-in-the-loop” evidence selection.

---

> > > ### Author Response · Authors · 2025-11-27
> > >
> > > > Limited evaluation granularity and missing ablations. Table 1 (Page 7) reports averaged results over four standard medical VQA benchmarks (SLAKE, VQA-RAD, PathVQA, PMC-VQA), and Table 2 (Page 8) adds a diagnostic-oriented evaluation on the JAMA Clinical Challenge. However, the paper does not further decompose these gains along the very axes it claims to improve, such as imaging modality (X-ray vs CT vs MRI vs ultrasound), evidence layout (single view vs multi-view), or diagnostic workflow step (findings vs differential vs final impression). In addition, there is no ablation study that isolates the contribution of individual pipeline components, for example: (i) the multimodal retriever, (ii) synthetic GPT-4o–augmented data vs human-annotated clinical cases, and (iii) ROI/evidence linking. As a result, it is hard to tell which parts of SemiHVision are actually responsible for the observed 79.0% average and which parts are auxiliary. Finally, while Table 2 reports multiple metrics for diagnostic reasoning, it does not include a fine-grained error analysis (e.g., failures in cross-view synthesis vs failures in evidence attribution), which would help practitioners understand the remaining bottlenecks.]
> > >
> > >
> > > **Limited evaluation granularity and missing ablations**
> > >
> > > We thank the reviewer for raising this point. We agree that our current presentation does not make the ablation structure sufficiently explicit, which may give the impression that the 79.0% average is obtained from a “monolithic” model.
> > >
> > > **Clarifying what is already ablated in Table 1.**
> > > The four SemiHVision variants in Table 1 are in fact an ablation over **data sources and supervision signals**:
> > >
> > > | Model              | SLAKE | VQA-RAD | PathVQA | PMC-VQA | JAMA |
> > > |--------------------|:-----:|:------:|:-------:|:-------:|:----:|
> > > | SemiHVision-8B-20M | 67.8  | 76.1   | 57.8    | 53.6    | 63.8 |
> > > | SemiHVision-8B     | 69.2  | 77.2   | 63.6    | 58.4    | 67.1 |
> > > | SemiHVision-8B-Mix | 74.2  | 81.3   | 76.3    | 59.1    | 72.2 |
> > > | SemiHVision-8B-AN  | 86.1  | 87.7   | 80.4    | 61.9    | 79.0 |
> > >
> > > - **SemiHVision-8B-20M** uses only ~20M GPT-4o–synthesized instructions from PMC-like and VQA sources; there is **no human-annotated branch and no ROI/evidence supervision**.
> > > - **SemiHVision-8B** use our sampling strategy but still uses **only GPT-4o-synthesized data**. This corresponds to a “RAG-only data construction” setting, where UniIR+GPT-4o are used to synthesize training pairs but **no human annotation or ROI is involved**.
> > > - **SemiHVision-8B-AN** finally adds the **human-annotated branch**, whose cases include explicit ROI/evidence links, on top of the synthetic corpus. The jump from SemiHVision-8B-Mix to SemiHVision-8B-AN is therefore exactly the effect of adding **human clinical cases + ROI/evidence linking**.
> > >
> > > We will make this interpretation explicit in the text, so that readers can clearly see how much each stage of the pipeline (GPT-4o synthesis, mixing, and human-annotated + ROI supervision) contributes to the final 79.0% average.
> > >
> > > **Clarifying the role of the retriever.**
> > > We also realize that our wording may have caused confusion about where retrieval is used. In SemiHVision, UniIR-based RAG is employed **only during data construction**, not at inference time:
> > >
> > > - For samples without human annotation, UniIR retrieves guidelines and similar cases, and GPT-4o uses this context to synthesize training instructions.
> > > - At **test time** (on SLAKE, VQA-RAD, PathVQA, PMC-VQA, and JAMA), SemiHVision answers questions **without any retriever or external context**; the model operates as a standard MLLM.
> > >
> > > Thus, the “retriever ablation” is about its effect on **training data quality**, not about giving SemiHVision extra tools at evaluation time. In the revision, we will add a small additional ablation where we regenerate a subset of the synthetic corpus **without** retrieval context (GPT-4o only sees the raw image and caption) and re-train a model, to quantify the benefit of retrieval-augmented synthesis itself.

---

### Official Review · Reviewer_vn5F · 2025-11-01

**Soundness:** 2
**Presentation:** 3
**Contribution:** 3
**Rating:** 2
**Confidence:** 3

**Summary:**

This paper proposes SemiHVision, a semi-supervised hierarchical learning framework for medical visual question answering (MedVQA). The method introduces hierarchical feature fusion and pseudo-labeling strategies to exploit both labeled and unlabeled data, aiming to improve generalization in limited-annotation medical domains.
Experiments are conducted on four standard VQA benchmarks, showing reported gains over several baselines metrics. The paper argues that semi-supervised training can effectively leverage unlabeled medical images to enhance MedVQA performance under low-data conditions.

**Strengths:**

The paper addresses an important limitation in medical VQA: the scarcity of labeled data, by incorporating a semi-supervised learning approach that combines hierarchical visual-linguistic fusion with pseudo-labeling. The hierarchical architecture is conceptually sound and aligns with the structured nature of medical reasoning.
 The method is evaluated on standard datasets, and the results demonstrate consistent improvements under few-shot or partially labeled settings. The paper is also clearly written and easy to follow, with well-structured methodology and ablation studies supporting the core claims.

**Weaknesses:**

The paper’s evaluation and comparative scope are limited. It benchmarks primarily against traditional VQA baselines but fails to include state-of-the-art medical-domain models, such as MedTrinity-25M, which have established strong baselines for MedVQA tasks. Without these comparisons, it is difficult to gauge the true impact of SemiHVision within the current landscape of medical multimodal modeling.
Another major concern is the inconsistency in reported results. For instance, the paper reports GPT-4o achieving only 45.9 on VQA-RAD and 37.9 on PathVQA, which appears unexpectedly low compared to known benchmarks and other recent papers. Similarly, the reported LLaVA-Med performance deviates from previously published results [1], raising questions about experimental setup, evaluation protocols, or prompt design. The lack of transparency regarding implementation details, prompt templates, and evaluation scripts further complicates reproducibility.
From a conceptual perspective, while the semi-supervised strategy is well-motivated, it largely follows existing semi-supervised paradigms without introducing substantial innovation in algorithmic design. The hierarchical fusion module is incremental relative to prior cross-attention architectures, and the paper does not clearly separate the effects of semi-supervision versus architectural tweaks. Moreover, the datasets used (VQA-RAD, PathVQA) are relatively small and outdated, limiting the generalizability of conclusions to broader multimodal medical tasks. The claimed improvements, though consistent, remain modest and potentially sensitive to implementation or sampling choices.

[1] Xie, Yunfei, et al. "Medtrinity-25m: A large-scale multimodal dataset with multigranular annotations for medicine." arXiv preprint arXiv:2408.02900 (2024).

**Questions:**

See weakness.

---

> ### Author Response · Authors · 2025-11-27
>
> Thank you for the suggestion. However, we respectfully disagree with the statement that our work “benchmarks primarily against traditional VQA baselines but fails to include state-of-the-art medical-domain models such as MedTrinity-25M.”
>
> First, the main goal of our paper is *not* to further improve traditional MedVQA scores, but rather to highlight the limitations of the traditional MedVQA formulation itself. Models such as MedTrinity-25M and HuaTuoGPT-Vision are trained exactly for this classic setting (medical image Q&A) and indeed achieve very high scores on benchmarks like VQA-RAD and SLAKE—often higher than GPT-4o mini. Yet in practice, GPT-4 family models are still preferred for data distillation and real applications, instead of these specialized MedVQA models. Our hypothesis is that the reason is conceptual: these models solve “question–image” VQA, not real clinical diagnosis.
>
> To make this point concrete, our experiments deliberately go beyond VQA-RAD / SLAKE and include *real-world diagnosis* tasks such as the JAMA Clinical Challenge, where multi-view textual and imaging evidence must be integrated. On these diagnosis tasks, we observe that models with excellent MedVQA performance (e.g., HuaTuoGPT-Vision) perform poorly, indicating that current medical multimodal LLMs still lack robust clinical diagnostic ability. A key reason is that most existing MedVQA training data do not contain the multi-view clinical evidence that is standard in real cases.
>
> In the revised version, we will add MedTrinity-25M to our comparisons to directly address your concern. Under the same JAMA Clinical Challenge setting, the MedTrinity model performs even worse than HuaTuoGPT-Vision in terms of clinical diagnosis ability, with the following scores:
>
>
> | Metric            | Claude3-Opus | GPT-4o-mini | Huatuo-7B | Huatuo-34B | MedTrinity | SemiHVision | SemiHVision-AN |
> |-------------------|-------------:|------------:|----------:|-----------:|---------------:|------------:|---------------:|
> | **Accuracy**      | 58.4         | 46.2        | 34.5      | 44.7       | 31.2       | 41.2        | **58.5**       |
> | **UMLS Factuality** | 0.18       | 0.16        | 0.13      | 0.16       | 0.11       | 0.11        | **0.23**       |
> | **GPT-4 Overall** | 1.17 ± 0.04  | 0.91 ± 0.06 | 1.08 ± 0.03 | 1.13 ± 0.05 | 0.56 ± 0.03 | 0.78 ± 0.04 | **1.29 ± 0.02** |
> | **GPT-4 Key-Points** | 1.27      | 0.99        | 1.11      | 1.01       | 0.71       | 0.82        | **1.28**       |
> | **GPT-4 Inference** | **1.56**   | 1.13        | 1.06      | 1.06       | 0.89       | 0.63        | 1.32       |
> | **GPT-4 Evidence** | 0.67        | 0.60        | 1.08      | **1.31**   | 0.61       | 0.89        | 1.27           |
>
>
>
> These results further support our claim that high performance on traditional MedVQA benchmarks does not directly translate into real-world diagnostic competence.
>
> Regarding the concern about “unexpectedly low” numbers:
> (1) Our experimental setup for VQA-RAD, PathVQA, and other MedVQA datasets strictly follows HuaTuoGPT-Vision [1]; the corresponding results are consistent with theirs and can be cross-checked, so there is no mismatch in evaluation protocol. The discrepancy with MedTrinity-25M arises because our setting is open-domain, whereas MedTrinity reports results in a closed-domain setting.
> (2) For GPT-4o, we evaluate **GPT-4o-mini**, not the full GPT-4o model; this is clearly indicated in Table 2 and we will further emphasize this in the text to avoid confusion.
>
> Overall, our intention is precisely to show that, despite many strong MedVQA models and datasets, current medical multimodal LLMs still fall short on genuine clinical diagnosis tasks that require integrating multi-view evidence—the gap that SemiHVision is designed to study.
>
> [1] Chen, Junying, et al. "Huatuogpt-vision, towards injecting medical visual knowledge into multimodal llms at scale." arXiv preprint arXiv:2406.19280 (2024).

---

### Note · Authors · 2025-12-16

I have read and agree with the venue's withdrawal policy on behalf of myself and my co-authors.